# Carbon and nitrogen contents in particle-size fractions of topsoil along a 3000 km aridity gradient in grasslands of northern China

X. G. Wang [1,2], S. A. Sistla [3], X. B. Wang [1], X. T. Lü [1*], X. G. Han [1]

[1] *Institute of Applied Ecology, Chinese Academy of Sciences, Shenyang 110164, China*

5    [2] *College of Environment and Resources, Dalian Minzu University, Dalian 116600, China*

[3] *Hampshire College, School of Natural Science, Amherst MA 01002, USA*

\* *Complete correspondence address to which the proofs should be sent*:

Wenhua Road 72, Shenhe District, Shenyang 110016, China

*Email*: lvxiaotao@iae.ac.cn

10    *Tel*: +86 24 83970752

*Fax*: +86 24 83970300

**Abstract.**

Climate factors such as aridity significantly influence soil carbon (C) and nitrogen (N) stocks in terrestrial ecosystems. Further, soil texture plays an important role in driving changes of soil C and N contents at regional scale. However, it remains uncertain whether such changes are resulted from the variation of different soil particle-size factions and/or the C and N concentrations in those fractions. We examined the distribution of total C and N in both bulk soil and different soil particle-size fractions, including sand (53-2000 μm), silt (2-53 μm) and clay (<2 μm), along a 3000 km transect in arid and semi-arid grasslands of northern China. Across the whole transect, sand content was positively and silt content was negatively correlated with increasing aridity. Carbon content in bulk soils (0-10 cm) ranged from 4.36 to 46.16 Mg C ha$^{-1}$, while N content ranged from 0.22 to 4.28 Mg N ha$^{-1}$ across different sampling sites on the transect. The total C and N concentrations and contents in bulk soils as well as in the three particle-size fractions tended to be negatively correlated with aridity. The concentrations and contents of total C and N in bulk soils were positively correlated with silt and clay contents and negatively correlated with sand content. Positive correlations were observed between the concentrations and contents of C or N in bulk soils and the C or N concentrations in the three soil particle-size fractions. By characterizing such a large scale aridity gradient, our results highlight that aridity would decrease soil C and N contents both by favoring increased sand content and by decreasing C and N concentrations in all the three soil fractions. These patterns thus have significant implications for

understanding soil C and N sequestration under scenarios of increasing aridity in

global drylands that are predicted to occur this century.

**Keywords** soil particle-size fractionation, soil organic matter, temperate steppe, carbon sequestration, precipitation to evapotranspiration, drought

**1 Introduction**

Grasslands, which cover nearly 40% of the world's land area, store

approximately one-third of the total carbon (C) in terrestrial ecosystems, and more

than 70% of C in grassland ecosystems is stored in the top 1m soil layer (He et al.,

2012; White et al., 2000). Consequently, C turnover in grassland soils is considered

to be a critical component of the global C cycle (Fisher et al., 1994; Wang et al.,

2009). In China, grasslands account for 41.7% of terrestrial land area and are mainly

distributed in arid and semi-arid regions (NSBC, 2002). Carbon and nitrogen (N)

cycling in the grasslands of northern China are relatively sensitive to global change

factors, such as increases in drought, more extreme precipitation events, and global

warming (Wang et al., 2014; Song et al., 2012). However, this sensitivity may show

regional variations due to variability in both climate and soil characteristics. Better

understanding of the controls on regional variation of soil C and N stocks in Chinese

grasslands would facilitate projections of regional C and N cycling under global

change scenarios.

Soil C and N stocks in grassland ecosystems are closely correlated with climatic

conditions. In arid and semi-arid ecosystems, soil C and N stocks are positively

correlated with mean annual precipitation (MAP) at the regional scale (He et al.,

2014; Nichols, 1984). This positive relationship is driven by the fact that water availability is the dominant limiting factor for plant growth (and thus soil organic matter inputs) in these ecosystems. In contrast, mean annual temperature (MAT) is negatively correlated with soil C and N stocks, as higher temperature generally

enhances microbial decomposition more than detrital production (Homann et al., 2007; Miller et al., 2004; Schimel et al., 1994).

Aridity, which is intensified by decreasing MAP and increasing MAT, is projected to increase in drylands worldwide during this century (Dai, 2013; Delgado-Baquerizo et al., 2013); this change may significantly diminish soil C and

N stocks in those regions (Delgado-Baquerizo et al., 2013; Sanaullah et al., 2014). Variation in soil fraction composition and the C and N concentrations in different soil particle sizes may strongly influence the pattern of decreasing bulk soil C and N pools observed with increasing aridity (Amelung et al., 1998; He et al., 2014). However, compared with these well-described patterns of variation in soil C and N

stocks along climate and soil particle size gradients, the relative influence of these factors on terrestrial C and N pools under climate change scenarios such as increasing aridity is less clear (Delgado-Baquerizo et al., 2013). Previous studies found that soil C and N tends to decrease with increasing aridity (that is the degree of dryness of the climate at a given location) largely due to decreased primary

productivity, as well as other ecosystem processes, such as reduction of total plant cover, shifts of species composition, changes of litter quality, altered litter decomposition rates (Carrera and Bertiller, 2010; Delgado-Baquerizo et al., 2013;

Sanaullah et al., 2014).

Variation in soil particle-size fractions also exerts significant controls on the stock and turnover of soil organic matter (SOM) (Chen et al., 2010; Christensen, 2001; Qin et al., 2010) and increasing attention has focused upon the responses of C and N pools in different soil particle-size fractions to climate change (Amelung et al., 1998; He et al., 2014). Clay and silt fractions in soil usually have higher C and N concentrations and stocks than that of sand fraction, and thus soils with higher clay and silt contents generally have higher soil organic C (SOC) and N stocks (Amelung et al., 1998; Feller and Beare, 1997; Follett et al., 2012; Hassink, 1997). This pattern reflects that organic materials are preferably decayed from pools of coarse soil particles; these relatively C- and N-rich decomposition products tend to accumulate in finer clay and silt particles (Amelung et al., 1998). Moreover, clay and silt may physically protect organic materials from decomposition and promote the accumulation of recalcitrant material in the fine particle-size fractions of soils (Hassink, 1997; Zhao et al., 2006; Chen and Chiu, 2003).

To increase our understanding of the variation in both the components of different soil particle-size fractions and their C and N concentrations with increasing aridity at the regional scale, we collected soil samples from 58 sites along a 3000 km transect in northern China which covered a wide range of grassland ecosystems locating at the eastern end of the contiguous Eurasian steppe with distinct aridity gradients. The objectives of this study were to: (1) examine the distribution of C and N in various particle-size fractions of soils across the aridity gradient, and (2)

100     evaluate the relative contributions of soil fraction components and their element

concentrations to the changes of soil C and N stocks across the aridity gradient. We

hypothesized that: (1) concentrations and stocks of C and N in soil particle-size

fractions would be negatively correlated with increasing aridity; and (2) soil C and N

105     stocks would decline with increasing aridity due to both an increase in the relative

proportion of sand to silt and clay and a decrease of C and N concentrations in the

three soil fractions.

## 2 Materials and methods

### 2.1 Study site and soil sampling

This study was carried out along a 3000 km west-east transect of arid and semi-arid grasslands across Xinjiang, Gansu, and Inner Mongolian in northern China. The distinctive features of this transect include complete meteorological records, relatively gentle geographical relief, and light human disturbances (Luo et al., 2015). The longitude of the transect ranged from 87°22′E to 123°23′E and the latitude ranged from 39°51′N to 50°3′N. The region is characterized by typical continental climate with limited precipitation occurring mainly in summer (Wang et al. 2015). Along the transect, the MAP increased from 34 mm in west region to 436 mm in east region, whereas the MAT ranged from -5 to 11℃. The aridity (calculated as "1－precipitation／evapotranspiration") of this transect ranged from 0.43 to 0.97 and derived from the WorldClim data set (http://www.worldclim.org/) (Hijmans et al., 2005). The main vegetation types were desert steppe, typical steppe, and meadow steppe from the west to east across the transect, with the primary productivity ranging from <10 g m$^{-2}$ yr$^{-1}$ in the desert steppe to >400 g m$^{-2}$ yr$^{-1}$ in the meadow steppe, and showing a negative relationship with aridity (Wang et al. 2014). The species richness per square meter ranged from 0 (no plants) in the west of the transect to >30 in the east parts (Lü et al. unpublished data). The soil types were mainly gray-brown desert soil, brown calcic soil, and chestnut soil (Kastanozem soil group) distributed along the transect from west to east (Luo et al. 2016). The large range of aridity gradients and these distinctive features of this area are facilitated to

study the relationships between C and N in grassland soils and climate change.

Fifty-eight sites were set up along the transect at an interval of 50-100 km. The location and elevation of the sampling sites were measured by GPS (eTrex Venture, Garmin, USA). For each site, one large plot (50 m×50 m) was selected and five subplots (1 m×1 m) were designated within the large plot (the four corners and the

center of the large plot). In each subplot, five soil samples (0-10 cm) were randomly collected by a 3.0 cm diameter soil corer and then totally mixed them together as one composite sample which was then sieved through a 2.0 mm sieve. All of the soil samples were returned to the lab and then air dried for further analysis. Soil bulk density (BD) was calculated as the ratio of dry soil mass per unit volume of the

sampling core (five cores were sampled for each site) and expressed as g cm$^{-3}$ (Grossman and Reinsch, 2002).

**2.2 Particle-size fractionation**

Particle-size fractionation was completed by disrupting soil aggregates of bulk soil samples using ultrasonic energy and separating the particle-size fractions by a

combination of wet sieving and continuous flow centrifugation (Chen and Chiu, 2003; He et al., 2009). Briefly, 40 g of sieved soil (< 2 mm) was dispersed in 200 ml of deionized water (the floating visible debris was removed) using a probe-type ultrasonic cell disrupter system (scientz-IID) operating for 15 min in the continuous mode at 361 W. We used a sieve to separate sand (particle size, 53-2000 μm) by

manual wet sieving method with deionized water. Particles which consisted of silt (2-53 μm) and clay (< 2 μm) passing through the sieve during the wet sieving

process were collected. In order to separate slit from clay, the mixture of particles and water was poured into a 500 ml centrifuge bottles and centrifuged at 682 rpm for 5 min. During this procedure, only the silt fraction sinks to the bottom while the clay fraction remains suspended. The silt fraction was then re-suspended in 200 ml deionized water and re-centrifuged at 476 rpm; this procedure was repeated 5 times. The clay fraction was obtained by transferring the suspensions into new centrifuge bottles and centrifuging them at 4000 rpm for 30 minutes. All the fractions were dried at 50 ℃ and then ground for further chemical analysis. The concentrations of total C and N in the bulk soil and soil particle-size fractions were determined using an automatic element analyzer (Vario MACRO cube, Elementar Analysensysteme GmbH).

**2.3 Calculations and statistical analysis**

Carbon and N stocks (Mg C ha$^{-1}$ and Mg N ha$^{-1}$, respectively) in bulk soils (0-10 cm) were calculated as follows:

C stocks $= D \times B \times C \times 100$

N stocks $= D \times B \times N \times 100$

where $D$, $B$, $C$, and $N$ represent the soil thickness (cm), BD (g cm$^{-3}$), C content (%), and N content (%), respectively.

Similarly, C and N stocks (Mg C ha$^{-1}$ and Mg N ha$^{-1}$, respectively) in sand, silt, and clay were calculated as follows:

C storage (fraction$_i$) $= D \times B \times C$ (fraction$_i$) $\times F_i \times 100$

N storage (fraction$_i$) $= D \times B \times N$ (fraction$_i$) $\times F_i \times 100$

where $C$ (fraction$_i$) is the C content of the soil fraction (%); $N$ (fraction$_i$) is the N

175    content of soil fraction (%); $F_i$ is the content of the fraction in the soil (%).

All of the relationships between variables were explored by using simple linear

regression analyses (58 sites with five subplots as replications in each site). We

observed that the relationships were best-fitted by either a first-order equation or a

second-order equation. As the contents of sand, silt and clay in soils are not

180    independent of each other, stepwise multiple regression analyses, which are highly

conservative (Fornara and Tilman, 2008), were used to determine the simultaneous

effects of soil fraction composition and C and N concentrations in soil particle-size

fractions on soil C and N stocks. All analyses were performed using SPSS V13.0

(SPSS, Chicago, IL, USA).

185

## 3 Results

### 3.1 Soil particle-size fractions and BD across the aridity transect

Sand was the most abundant fraction for most sites, accounting for 21.62-90.65% of the total soil weight along the transect. The content of sand was positively correlated with increasing aridity (Fig. 1). The silt content, which accounted for 4.19-49.29% of the total soil weight, decreased with increasing aridity (Fig. 1). The content of clay was relatively low across the transect, ranging from 1.36-33.7%. There were no significant relationships between clay content and aridity (Fig. 1). Bulk density ranged from 0.90 to 1.72 g cm$^{-3}$ and was positively correlated with aridity.

### 3.2 C and N concentrations in bulk soil and different soil particle-size fractions

Total C (Fig. 2a) and N concentrations (Fig. 2b) in the bulk soil significantly decreased with increasing aridity. Soil C concentration ranged from 2.71 to 50.33 g C kg$^{-1}$, while the N concentration ranged from 0.14 to 4.75 g N kg$^{-1}$ (Table 1). Across the transect, C and N concentrations in all of the three particle-size fractions were negatively correlated with increasing aridity (Fig. 2). The total C and N concentrations in the soil particle-size fractions varied greatly among the three soil fractions (Table 1), with the highest concentrations in clay (36.06±1.49 g C kg$^{-1}$ and 3.90±0.17 g N kg$^{-1}$, respectively) and the lowest in sand (5.19±0.56 g C kg$^{-1}$ and 0.37±0.04 g N kg$^{-1}$, respectively) ($p < 0.001$ in both cases).

### 3.3 C and N stocks in bulk soil and different soil particle-size fractions

Across the whole transect, C stock in bulk soils (0-10 cm) ranged from 4.36 to

46.16 Mg C ha$^{-1}$, while N stock ranged from 0.22 to 4.28 Mg N ha$^{-1}$ (Table 2). We

found negative correlations between C and N stocks in each soil particle-size

fraction    and    aridity,    with    the    exception    of    C    stocks    in    sand

($y$=57.12-161.29$x$+114.70$x^2$, $R^2$=0.48, $p$<0.0001), which first decreased and then

increased with increasing aridity (Fig. 3a). Paralleling this pattern, C stocks in bulk

soils first decreased and then slightly increased with increasing aridity, with the

lowest value presented in sites with aridity of ~0.8 ($y$=160.49-371.88$x$+228.73$x^2$,

$R^2$=0.73, $p$<0.0001, Fig. 3a). Nitrogen stocks in bulk soils were negatively correlated

with aridity (Fig. 3b).

**3.4 Relationships between soil fraction composition, C and N concentrations in**

**soil particle-size fractions and bulk soil C and N stocks**

Across the transect, the concentrations and stocks of C and N in bulk soils were

negatively correlated with the content of sand and positively correlated with the

contents of silt and clay (Fig. 4). The concentrations and stocks of C and N in bulk

soils were positively correlated with their concentrations in sand, silt, and clay (Fig.

5).

Stepwise multiple regression analyses allowed us to quantify the simultaneous

effects of soil fraction composition, element concentrations in soil particle-size

fractions, and BD on bulk soil C and N stocks. The multiple regression model for C

stocks in bulk soils included the variables (Table 3): clay C concentration (with the

value of normalized regression coefficient for this variable = 0.70), clay content (0.45), silt content (0.21) and silt C concentration (-0.12), while sand content, sand C concentration and BD were excluded from the model. These variables together accounted for 93.8% of the total variation of bulk soil C stock. In contrast, the

multiple regression model that best predicted soil N stock included (Table 4): clay N concentration (0.61), sand content (-0.30), clay content (0.27), BD (-0.10), and sand N concentration (-0.05), while silt content and silt N concentration were excluded from the model. These variables accounted for 93.6% of the total variation of bulk soil N stock. Inconsistent with the results of simple linear regressions, the stepwise

multiple regression analyses showed that C concentration in silt had a negative correlation with soil C stock, and that sand N concentration had a negative correlation with soil N stock.

**4 Discussion**

245   Across this 3000 km aridity gradient, sand, which accounted for 21.62-90.65% of the total soil weight, was the most abundant fraction; the contents of silt (4.19-49.29%) and clay (1.36-33.7%) were much lower, especially in soils from the extremely arid sites. We suspect that this pattern is partly caused by the wind erosion and dust storms which can be exacerbated by increasing aridity and frequently occur

250 in higher aridity areas of northern China (Wang et al., 2013; Yan et al., 2013; Zhang and Liu, 2010). Wind erosion favors losses of fine soil particles and consequently leads to changes of the soil texture (Feng et al., 2001; Wang et al., 2006; Yan et al., 2013). In arid and semi-arid ecosystems experiencing increasing aridity, soils become more vulnerable to wind erosion because vegetation coverage declines

255 (Zhang and Liu, 2010). Similar to our results, Liu et al. (2008) found that sand fractions in soils of steppe and meadow were negatively correlated with MAP and positively correlated with MAT due to drought-driven vegetation cover decline in the semi-arid East Asian steppe.

   Other factors may also contribute to the pattern observed here. Parent materials,

260 land use (e.g. grazing), and topography, can largely influence soil formation process and the contents of soil fractions (Barthold et al., 2013; Deng et al., 2015). For example, soils derived from limestone and quartzite have a lower content of sand fractions and a higher content of silt, compared to soils derived from granite (Belnap et al., 2014). In the grasslands of Inner Mongolia, climate and land use (e.g.

265 intensified grazing leads to soil degradation by diminishing the fine soil fraction) are

of greater importance than parent material and topography in controlling soil type distribution (Barthold et al., 2013). Therefore, we suspect that those factors associated with climate would be more important than other factors in structuring the pattern of soil particle-size distribution in our study sites.

Our results showed that C and N concentrations were highest in clay, followed by silt, and much lower in sand across a 3000 km aridity gradient. This pattern, which may be caused because fine fractions in soil have high surface area which can enhance formation of organo-mineral complexes that protect SOM from microbial degradation (Hassink, 1997; Zhang and Liu, 2010), supports previous findings that

soil fractionation is a useful tool for examining different C and N pools in soil (Amelung et al., 1998; Gerzabek et al., 2001; Stemmer et al., 1999). Across the transect, we observed that C and N concentrations and stocks in bulk soils were negatively correlated with sand content and positively correlated with silt and clay contents. Similarly, Bai et al. (2007) demonstrated that there was a negative

correlation between SOC content and sand content, and there were positive correlations between SOC content and clay and silt contents based in wetland soils in northeastern China. Positive correlations between soil C and N concentrations and silt and clay contents were also found in Inner Mongolian grasslands (He et al., 2014).

Supporting our first hypothesis, we found that C and N concentrations and stocks in soil particle-size fractions tended to be negatively correlated with increasing aridity. The higher aridity sites have lower primary productivity (Wang et

al., 2014) and thus a lower input of plant detritus into soil. Lower litter input is correlated with lower C and N concentrations and stocks in soil fractions (Yang et al., 2011; He et al., 2014). Additionally, aridity has been identified to be a major factor affecting bacterial diversity, community composition and taxon abundance in this system (Wang et al. 2015). Therefore, with increasing aridity, microbially-mediated litter decomposition may also change due to altered microbial community composition, which may further influence soil C and N (Carrera and Bertiller, 2010). Paralleling our results, He et al. (2014) found that C and N concentrations in soil particle-size fractions were positively correlated with MAP; moreover, they considered that MAP was better than MAT to model the variation of soil C stock in an Inner Mongolia grassland. In the present study, we quantified the relationship between soil C and N stocks and aridity, which combines MAP and MAT, across different sampling sites. Our results suggest that aridity is a robust predictor for the regional variation of C and N stocks in soil fractions.

We found that C stock in sand was first decreased and then increased along the aridity gradient, which seems paradoxical given the results that the C concentrations in sand linearly declined with increasing aridity while the content of sand linearly increased with increasing aridity across the transect. The observed variation of C stock in sand across the transect may be due to the shifts of dominant controller for the C stock in sand across the aridity transect. Sand C concentration appears to be more important than sand content in driving the variation of sand C stock in the ecosystem with aridity value is less than 0.8 (where the C concentration in sand was

relatively higher and sand content was relatively lower). In contrast, sand content

appears to be more important than C concentration in determining sand C stock

when the aridity value exceeds 0.8. Our results highlight the importance of

considering both soil particle size and the C concentration of different particles in

order to better understand the influence of aridity on soil C pools.

We found that total C and N concentrations and stocks in bulk soils generally

decreased with increasing aridity across the whole transect. Previous studies have

reported that soil C and N stocks in the upper soil layers were positively correlated

with MAP and negatively correlated with MAT; these findings are similar to our

observations along a large aridity gradient (Follett et al., 2012; He et al., 2014; Liu et

al., 2012; Miller et al., 2004). The depletion of fine soil particles due to the

intensified wind erosion with increasing aridity could further deplete C and nutrients

in arid systems because these particles have disproportionately greater amounts of C

and nutrients than larger particles (Yan et al., 2013). Furthermore, the decline of

plant coverage and aboveground biomass under higher aridity would also contribute

to the decreased C and N content along this aridity gradient. Actually, aboveground

primary productivity was significantly decreased with increasing aridity along this

transect (Wang et al., 2014).

Our results suggest that the decreases of bulk soil C and N stocks along the

aridity gradient were resulted not only from the changes of composition of different

soil fractions but also from the decreases of C and N concentrations in each of those

fractions. While both soil C and N stocks decreased with increasing aridity, the

stepwise multiple regression analyses indicated that the simultaneous influences of variation of different soil fractions and the element concentrations were different for C and N. For bulk soil C stock, the most robust regression model did not include sand content, sand C concentration, and BD, whereas for bulk soil N stock, silt content and silt N concentration were excluded from the model. Our results thus demonstrate that sand content is less important than silt content for controlling variation of soil C stock, whereas silt is less important for the variation of soil N stock at regional scale in the arid and semi-arid grasslands of northern China.

These findings are somewhat in agreement with previous findings that C is readily mineralized from un-complexed organic matter in sand-sized aggregates whereas N is not, while silt tends to be more enriched in C than N (Christensen, 2001). We found that clay content and clay element concentrations were the most important factors for predicting the variation of both the soil C and N stocks across this aridity gradient. Similarly, Burke et al. (1989) observed that clay was an important predictor of soil C for American grassland soils. Together, these results indicate differences in the relative importance of different soil particle-size fractions in driving soil C and N stocks, although it is generally accepted that the dynamics of those two elements in soils are closely correlated (Finzi et al., 2011).

This large-scale field investigation provides strong evidence that increasing aridity would reduce the soil C and N stocks in arid and semi-arid ecosystems due both to the changes of particle-sized fractions in soils (i.e. relatively more coarse fraction content, but less fine fraction content with increasing aridity) and to the

decline of C and N concentrations in each fraction. This study provides novel

insights into the patterns underlying regional changes of soil C and N from a soil

particle-size fractions perspective. Given the predicted increases in aridity in this

century for the global drylands (Dai, 2013), this study indicates that the soil C and N

pools in those arid ecosystems may decline in the future. Because wind erosion

would lead to greater loss of relatively fine silt and clay particles (Yan et al., 2013),

our results suggest that land use practices which reduce wind erosion (e.g. reducing

the intensity of grazing) will play an important role in sustaining soil C sequestration

in dryland regions globally.

**5 Conclusions**

Along the transect, aridity was an important factor driving the changes of soil C and N concentrations and stocks in the arid and semi-arid grasslands. Both of the C and N concentrations and stocks in the three particle-size fractions as well as in bulk soils tended to be negatively correlated with aridity. The concentrations and stocks of C and N in bulk soils were negatively correlated with sand content but positively

correlated with both silt and clay contents, suggesting that fine soil fractions can protect SOM from microbial degradation. There were positive correlations between the concentrations and stocks of C or N in bulk soils and the C or N concentrations in the three soil particle-size fractions. Our results have significant implications for better understanding soil C and N cycles under scenarios of increasing aridity in

global drylands that are predicted to occur this century.

**Acknowledgements**

We thank all the members in the Shenyang Sampling Campaign Team from the Institute of Applied Ecology, Chinese Academy of Sciences for their assistance in field sampling. This work was supported by National Natural Science Foundation of China (31470505), the National Basic Research Program of China (2015CB150802), Strategic Priority Research Program of the Chinese Academy of Sciences (XDB15010403 and XDB15010401), the Key Research Program from CAS (KFZD-SW-305-002) and Youth Innovation Promotion Association CAS (2014174).

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

Table legend

**Table 1** Soil C and N concentrations in bulk soils and different soil particle-size fractions (sand, silt and clay) at 58 sampling sites in arid and semi-arid grasslands of northern China (Data are represented as means $\pm 1$ SE, n=5)

**Table 2** Soil C and N stocks in bulk soils and different soil particle-size fractions (sand, silt and clay) at 58 sampling sites in arid and semi-arid grasslands of northern China (Data are represented as means $\pm 1$ SE, n=5)

**Table 3** Results of the multiple regressions refer to the final accepted model which just included

the effects of the significant variables for C stocks in bulk soils of arid and semi-arid grasslands

**Table 4** Results of the multiple regressions refer to the final accepted model which just included the effects of the significant variables for N stocks in bulk soils of arid and semi-arid grasslands

Figure legend

**Fig. 1** The relationships of soil particle-size fractions contents with aridity differed, with positive

correlation between sand content and aridity, negative correlation between silt content and aridity,

and no significant relationship between clay content and aridity. Data are presented as mean±1SE

(n=5)


**Fig. 2** Carbon (a) and N concentrations (b) in all of the three particle-size fractions, as well as in

bulk soils, were negatively correlated with increasing aridity. Data are presented as mean±1SE

(n=5)

**Fig. 3** The C (a) and N stocks (b) in bulk soil and soil particle-size fractions were generally

negatively correlated with aridity. Data are presented as mean±1SE (n=5)

**Fig. 4** Across the transect, the concentrations and stocks of C (a, b) and N (c, d) in bulk soils

were negatively correlated with the content of sand and positively correlated with the contents of

silt and clay. Data are presented as mean±1SE (n=5)

**Fig. 5** The concentrations and stocks of C (a, b) and N (c, d) in bulk soils were positively

correlated with their concentrations in sand, silt, and clay. Data are presented as mean±1SE (n=5)

Table 1

| site | | C concentrations (g C kg$^{-1}$) | | | | N concentrations (g N kg$^{-1}$) | | | |
|---|---|---|---|---|---|---|---|---|---|
| latitude | longitude | bulk soil | sand | silt | clay | bulk soil | sand | silt | clay |
| 42°13′24.02″ | 87°22′37.83″ | 8.66±0.70 | 3.28±0.37 | 28.71±1.62 | 36.52±6.17 | 0.32±0.05 | 0.11±0.01 | 0.67±0.13 | 3.02±0.65 |
| 42°59′17.72″ | 90°25′32.02″ | 9.69±0.49 | 7.08±0.05 | 19.3±2.07 | 12.28±2.34 | 0.49±0.04 | 0.33±0.02 | 0.45±0.03 | 0.78±0.10 |
| 43°16′04.17″ | 91°15′35.87″ | 7.54±1.13 | 4.71±0.34 | 21.90±1.13 | 17.13±2.05 | 0.81±0.48 | 0.17±0.02 | 0.56±0.10 | 1.38±0.36 |
| 43°24′13.23″ | 91°54′43.13″ | 8.35±1.01 | 5.05±0.14 | 21.69±1.79 | 26.68±4.62 | 0.47±0.12 | 0.20±0.02 | 0.80±0.16 | 2.43±0.52 |
| 43°07′38.09″ | 92°48′45.88″ | 11.5±0.58 | 6.19±0.34 | 33.21±1.13 | 44.71±3.31 | 0.58±0.09 | 0.22±0.03 | 0.97±0.12 | 3.05±0.38 |
| 42°58′04.74″ | 93°27′34.27″ | 6.14±0.46 | 3.43±0.21 | 20.86±0.57 | 23.34±1.05 | 0.38±0.04 | 0.25±0.04 | 0.99±0.08 | 2.10±0.12 |
| 42°41′55.57″ | 93°58′53.51″ | 3.85±0.15 | 1.75±0.09 | 18.21±1.92 | 19.11±2.39 | 0.36±0.15 | 0.15±0.03 | 0.65±0.21 | 1.45±0.36 |
| 42°15′37.46″ | 94°16′44.10″ | 12.97±2.27 | 10.03±1.46 | 30.65±5.63 | 8.96±0.65 | 0.14±0.01 | 0.17±0.01 | 0.28±0.02 | 0.73±0.08 |
| 41°34′12.85″ | 95°17′50.04″ | 9.78±0.80 | 5.62±0.68 | 24.42±0.69 | 18.45±1.34 | 0.33±0.02 | 0.19±0.02 | 0.55±0.04 | 1.39±0.16 |
| 39°51′53.47″ | 98°39′20.30″ | 13.71±0.82 | 11.94±0.80 | 29.24±4.11 | 13.38±1.24 | 0.28±0.01 | 0.14±0.03 | 0.49±0.03 | 0.88±0.07 |
| 40°14′41.91″ | 99°20′35.82″ | 14.38±0.75 | 14.25±0.78 | 21.10±0.50 | 16.44±2.13 | 0.20±0.03 | 0.19±0.02 | 0.55±0.01 | 1.01±0.05 |
| 40°28′46.39″ | 99°51′59.92″ | 7.73±0.18 | 6.48±0.23 | 26.67±1.71 | 8.36±0.40 | 0.26±0.01 | 0.16±0.03 | 0.42±0.03 | 0.87±0.06 |
| 41°08′18.31″ | 100°27′38.72″ | 8.10±0.52 | 5.13±0.18 | 21.76±1.54 | 15.72±2.09 | 0.36±0.05 | 0.17±0.02 | 0.46±0.04 | 1.25±0.19 |
| 41°39′47.22″ | 100°58′30.60″ | 8.83±0.34 | 6.80±0.21 | 22.43±0.56 | 10.93±1.20 | 0.32±0.02 | 0.18±0.01 | 0.41±0.06 | 0.66±0.04 |
| 42°00′49.37″ | 101°42′23.12″ | 6.48±0.27 | 6.40±0.26 | 10.01±0.74 | 6.01±0.29 | 0.20±0.02 | 0.19±0.03 | 0.47±0.05 | 0.75±0.06 |
| 41°57′49.08″ | 102°20′41.89″ | 3.95±0.22 | 2.55±0.13 | 22.12±1.11 | 9.64±1.40 | 0.19±0.03 | 0.19±0.01 | 0.45±0.04 | 0.91±0.06 |
| 41°43′02.80″ | 103°06′51.38″ | 9.19±0.24 | 6.99±0.40 | 21.62±1.17 | 11.04±0.98 | 0.35±0.04 | 0.23±0.02 | 0.53±0.03 | 0.74±0.02 |
| 41°21′26.33″ | 103°45′42.79″ | 4.20±0.23 | 3.25±0.17 | 17.01±0.55 | 5.60±0.42 | 0.18±0.01 | 0.19±0.01 | 0.47±0.05 | 0.63±0.06 |
| 40°52′33.64″ | 104°26′49.14″ | 9.01±0.52 | 4.29±0.35 | 22.52±1.58 | 22.91±0.97 | 0.31±0.03 | 0.16±0.02 | 0.68±0.04 | 1.50±0.13 |
| 40°47′37.30″ | 104°52′51.45″ | 5.32±0.95 | 1.77±0.11 | 19.27±2.01 | 21.95±2.22 | 0.23±0.03 | 0.16±0.02 | 0.53±0.07 | 1.13±0.14 |
| 40°43′44.48″ | 105°36′43.8″ | 4.48±0.54 | 2.60±0.43 | 14.61±0.64 | 14.71±0.88 | 0.22±0.03 | 0.21±0.03 | 0.67±0.03 | 1.64±0.08 |

| | | | | | | | | |
|---|---|---|---|---|---|---|---|---|
| 40°41′025″ | 106°02′886″ | 5.97 ±1.06 | 1.71 ±0.12 | 17.42 ±1.42 | 27.64 ±0.92 | 0.35 ±0.06 | 0.18 ±0.03 | 0.88 ±0.08 | 1.91 ±0.14 |
| 41°27′02.23″ | 107°00′06.04″ | 12.70 ±2.16 | 1.31 ±0.27 | 21.08 ±2.68 | 37.47 ±2.47 | 0.76 ±0.08 | 0.19 ±0.01 | 1.31 ±0.08 | 2.31 ±0.09 |
| 41°47′46.44″ | 107°28′11.62″ | 6.36 ±0.30 | 0.81 ±0.14 | 11.71 ±0.69 | 24.14 ±1.47 | 0.58 ±0.02 | 0.19 ±0.02 | 1.11 ±0.08 | 2.23 ±0.04 |
| 41°49′42.97″ | 107°36′40.26″ | 6.3 ±0.40 | 0.64 ±0.04 | 13.84 ±0.81 | 31.14 ±1.32 | 0.64 ±0.05 | 0.16 ±0.02 | 1.54 ±0.13 | 3.46 ±0.18 |
| 41°51′57.84″ | 108°03′14.43″ | 4.85 ±0.21 | 1.01 ±0.08 | 11.26 ±2.86 | 20.26 ±0.80 | 0.48 ±0.04 | 0.19 ±0.03 | 1.06 ±0.27 | 2.69 ±0.06 |
| 41°54′52.38″ | 108°42′38.69″ | 6.75 ±0.38 | 1.01 ±0.03 | 14.41 ±0.77 | 29.94 ±0.95 | 0.80 ±0.06 | 0.19 ±0.02 | 1.84 ±0.10 | 4.20 ±0.07 |
| 42°09′46.07″ | 109°09′56.15″ | 5.80 ±0.29 | 0.84 ±0.06 | 16.83 ±0.78 | 26.45 ±0.61 | 0.64 ±0.05 | 0.15 ±0.02 | 1.80 ±0.07 | 3.23 ±0.06 |
| 42°24′54.72″ | 109°48′18.06″ | 4.39 ±0.27 | 0.56 ±0.02 | 9.37 ±0.31 | 12.12 ±0.34 | 0.71 ±0.05 | 0.22 ±0.03 | 1.40 ±0.05 | 2.31 ±0.04 |
| 42°37′26.24″ | 110°17′41.72″ | 4.17 ±0.33 | 0.72 ±0.02 | 11.35 ±1.11 | 13.26 ±0.46 | 0.54 ±0.03 | 0.24 ±0.02 | 1.29 ±0.08 | 2.05 ±0.08 |
| 42°55′54.74″ | 110°49′27.19″ | 2.93 ±0.10 | 0.42 ±0.03 | 10.82 ±0.61 | 14.99 ±0.30 | 0.46 ±0.04 | 0.19 ±0.03 | 1.58 ±0.09 | 2.64 ±0.06 |
| 43°08′49.69″ | 111°21′18.26″ | 5.18 ±0.55 | 0.66 ±0.06 | 12.60 ±0.77 | 15.71 ±0.73 | 0.60 ±0.05 | 0.18 ±0.01 | 1.65 ±0.06 | 2.41 ±0.06 |
| 43°22′54.51″ | 111°57′51.32″ | 9.54 ±0.21 | 1.09 ±0.05 | 25.12 ±0.68 | 19.64 ±0.71 | 0.71 ±0.02 | 0.18 ±0.01 | 1.83 ±0.03 | 1.79 ±0.04 |
| 43°38′05.12″ | 112°11′47.03″ | 2.71 ±0.25 | 0.46 ±0.05 | 10.26 ±1.22 | 22.59 ±0.40 | 0.36 ±0.04 | 0.15 ±0.02 | 1.41 ±0.11 | 3.39 ±0.04 |
| 43°42′25.70″ | 112°55′16.72″ | 6.08 ±0.69 | 0.54 ±0.04 | 17.40 ±1.79 | 23.09 ±2.45 | 0.70 ±0.01 | 0.16 ±0.01 | 1.99 ±0.05 | 2.94 ±0.08 |
| 43°49′08.29″ | 113°27′58.82″ | 4.14 ±0.32 | 0.75 ±0.05 | 15.89 ±0.88 | 23.87 ±0.39 | 0.49 ±0.04 | 0.15 ±0.02 | 1.98 ±0.06 | 3.68 ±0.04 |
| 43°50′58.62″ | 114°05′08.22″ | 6.12 ±0.26 | 0.87 ±0.07 | 17.61 ±1.35 | 27.75 ±0.64 | 0.85 ±0.03 | 0.19 ±0.01 | 2.29 ±0.17 | 4.24 ±0.05 |
| 43°58′46.01″ | 114°49′36.29″ | 9.63 ±1.51 | 1.10 ±0.27 | 23.57 ±2.37 | 37.96 ±3.76 | 1.17 ±0.07 | 0.17 ±0.03 | 2.76 ±0.24 | 4.99 ±0.06 |
| 43°55′33.55″ | 115°42′06.75″ | 9.83 ±0.74 | 1.07 ±0.12 | 27.50 ±1.47 | 37.53 ±1.10 | 1.29 ±0.09 | 0.20 ±0.01 | 3.37 ±0.16 | 5.26 ±0.12 |
| 44°13′17.46″ | 116°30′25.43″ | 11.47 ±0.16 | 1.02 ±0.09 | 29.23 ±1.47 | 46.78 ±0.48 | 1.37 ±0.02 | 0.23 ±0.01 | 3.27 ±0.15 | 5.93 ±0.05 |
| 44°27′59.70″ | 117°10′47.04″ | 15.17 ±1.52 | 2.24 ±0.24 | 36.64 ±2.20 | 53.74 ±2.63 | 1.72 ±0.17 | 0.25 ±0.03 | 3.97 ±0.24 | 6.73 ±0.19 |
| 44°39′58.27″ | 117°53′44.73″ | 25.5 ±1.11 | 4.65 ±0.59 | 40.01 ±1.13 | 67.17 ±2.25 | 2.70 ±0.08 | 0.50 ±0.06 | 4.16 ±0.12 | 8.07 ±0.28 |
| 44°59′23.89″ | 118°44′44.89″ | 14.53 ±0.98 | 3.32 ±0.44 | 44.87 ±1.71 | 72.27 ±1.25 | 1.49 ±0.10 | 0.30 ±0.03 | 4.61 ±0.13 | 8.48 ±0.13 |
| 45°25′36.47″ | 119°43′23.02″ | 21.09 ±1.08 | 3.94 ±0.73 | 47.37 ±1.35 | 76.83 ±0.75 | 1.97 ±0.08 | 0.31 ±0.04 | 4.16 ±0.12 | 8.08 ±0.05 |
| 46°22′37.77″ | 120°28′38.48″ | 37.01 ±2.79 | 16.31 ±3.17 | 44.64 ±3.27 | 68.97 ±2.69 | 3.34 ±0.24 | 1.23 ±0.20 | 3.39 ±0.26 | 6.98 ±0.32 |
| 47°39′21.62″ | 119°17′57.40″ | 38.17 ±2.43 | 12.67 ±1.58 | 45.00 ±3.53 | 89.56 ±5.70 | 3.50 ±0.24 | 0.94 ±0.13 | 3.81 ±0.40 | 9.16 ±0.56 |

| | | | | | | | | |
|---|---|---|---|---|---|---|---|---|
| 48°05′19.80″ | 118°27′20.04″ | 17.26±0.54 | 2.33±0.30 | 36.19±0.95 | 64.34±1.13 | 1.76±0.04 | 0.24±0.03 | 3.41±0.10 | 7.30±0.10 |
| 48°20′40.36″ | 117°58′46.17″ | 9.31±0.60 | 1.08±0.11 | 28.08±1.75 | 50.39±2.86 | 1.03±0.07 | 0.22±0.03 | 2.79±0.17 | 6.09±0.28 |
| 48°29′493″ | 117°09′716″ | 6.88±0.18 | 1.11±0.08 | 37.61±1.67 | 61.59±1.78 | 0.87±0.02 | 0.21±0.02 | 3.77±0.10 | 7.47±0.14 |
| 48°51′26.99″ | 116°53′36.00″ | 9.06±0.92 | 1.04±0.18 | 27.98±2.37 | 40.5±1.90 | 1.13±0.09 | 0.18±0.03 | 2.81±0.24 | 4.97±0.33 |
| 49°20′17.60″ | 117°05′28.24″ | 12.27±1.37 | 2.58±0.42 | 63.65±4.30 | 75.87±2.33 | 1.35±0.13 | 0.30±0.06 | 5.89±0.41 | 8.81±0.22 |
| 49°31′45.39″ | 118°00′35.84″ | 9.16±0.23 | 1.23±0.06 | 60.19±1.77 | 80.9±1.33 | 1.05±0.01 | 0.17±0.02 | 5.59±0.14 | 9.40±0.16 |
| 49°47′01.88″ | 118°32′00.47″ | 13.21±0.58 | 2.72±0.31 | 65.95±2.72 | 75.77±1.76 | 1.50±0.06 | 0.32±0.04 | 6.13±0.20 | 8.93±0.25 |
| 50°03′13.52″ | 119°16′57.97″ | 19.96±1.08 | 3.63±0.14 | 61.42±2.35 | 75.72±2.00 | 2.05±0.08 | 0.34±0.01 | 5.38±0.15 | 8.18±0.15 |
| 49°52′44.41″ | 119°59′36.93″ | 48.93±1.05 | 37.62±1.73 | 43.74±2.02 | 62.92±2.50 | 4.35±0.06 | 2.78±0.18 | 3.50±0.13 | 6.11±0.23 |
| 49°28′48.71″ | 119°40′55.44″ | 50.33±3.21 | 59.71±5.17 | 37.20±4.97 | 53.87±3.69 | 4.75±0.30 | 4.81±0.41 | 3.17±0.42 | 5.51±0.39 |
| 49°11′195″ | 120°21′451″ | 28.07±3.15 | 5.72±1.02 | 56.70±4.65 | 100.73±5.35 | 2.64±0.28 | 0.53±0.08 | 5.02±0.43 | 10.33±0.51 |
| 44°46′16.79″ | 123°23′04.99″ | 9.17±0.43 | 1.48±0.28 | 19.78±4.43 | 59.23±2.83 | 0.87±0.13 | 0.23±0.03 | 1.91±0.46 | 5.57±0.94 |

Table 2

| site | | C stocks (Mg C ha$^{-1}$) | | | | N stocks (Mg N ha$^{-1}$) | | | |
| --- | --- | --- | --- | --- | --- | --- | --- | --- | --- |
| latitude | longitude | bulk soil | sand | silt | clay | bulk soil | sand | silt | clay |
| 42°13′24.02″ | 87°22′37.83″ | 13.93 ±1.12 | 4.32 ±0.50 | 6.96 ±0.87 | 1.47 ±0.21 | 0.52 ±0.08 | 0.15 ±0.02 | 0.15 ±0.02 | 0.12 ±0.02 |
| 42°59′17.72″ | 90°25′32.02″ | 15.71 ±0.79 | 8.07 ±0.31 | 7.19 ±0.77 | 1.18 ±0.40 | 0.79 ±0.07 | 0.37 ±0.02 | 0.17 ±0.02 | 0.08 ±0.02 |
| 43°16′04.17″ | 91°15′35.87″ | 12.3 ±1.84 | 6.29 ±0.53 | 4.47 ±1.37 | 1.23 ±0.42 | 1.32 ±0.79 | 0.23 ±0.04 | 0.11 ±0.04 | 0.09 ±0.02 |
| 43°24′13.23″ | 91°54′43.13″ | 13.43 ±1.63 | 6.96 ±0.18 | 3.82 ±0.96 | 1.61 ±0.43 | 0.76 ±0.19 | 0.28 ±0.03 | 0.15 ±0.05 | 0.15 ±0.05 |
| 43°07′38.09″ | 92°48′45.88″ | 18.4 ±0.92 | 8.37 ±0.51 | 5.72 ±0.46 | 3.40 ±0.37 | 0.92 ±0.14 | 0.30 ±0.04 | 0.16 ±0.02 | 0.23 ±0.03 |
| 42°58′04.74″ | 93°27′34.27″ | 9.85 ±0.74 | 4.80 ±0.37 | 3.07 ±0.67 | 1.41 ±0.21 | 0.61 ±0.06 | 0.34 ±0.05 | 0.14 ±0.03 | 0.13 ±0.02 |
| 42°41′55.57″ | 93°58′53.51″ | 6.24 ±0.24 | 2.46 ±0.17 | 2.82 ±0.47 | 1.04 ±0.18 | 0.59 ±0.24 | 0.22 ±0.05 | 0.10 ±0.03 | 0.08 ±0.02 |
| 42°15′37.46″ | 94°16′44.10″ | 21.16 ±3.70 | 12.80 ±1.67 | 8.71 ±2.43 | 0.67 ±0.17 | 0.22 ±0.02 | 0.21 ±0.02 | 0.08 ±0.01 | 0.05 ±0.01 |
| 41°34′12.85″ | 95°17′50.04″ | 15.86 ±1.30 | 6.67 ±0.73 | 7.60 ±0.33 | 2.08 ±0.36 | 0.54 ±0.03 | 0.22 ±0.02 | 0.17 ±0.01 | 0.16 ±0.04 |
| 39°51′53.47″ | 98°39′20.30″ | 22.19 ±1.33 | 15.15 ±1.17 | 4.86 ±0.95 | 2.54 ±0.45 | 0.46 ±0.02 | 0.17 ±0.04 | 0.08 ±0.00 | 0.16 ±0.01 |
| 40°14′41.91″ | 99°20′35.82″ | 23.41 ±1.21 | 20.91 ±0.57 | 2.82 ±0.97 | 0.34 ±0.04 | 0.33 ±0.04 | 0.28 ±0.03 | 0.07 ±0.02 | 0.02 ±0.00 |
| 40°28′46.39″ | 99°51′59.92″ | 12.61 ±0.29 | 9.25 ±0.45 | 2.95 ±0.34 | 0.78 ±0.08 | 0.42 ±0.02 | 0.23 ±0.04 | 0.05 ±0.01 | 0.08 ±0.01 |
| 41°08′18.31″ | 100°27′38.72″ | 13.21 ±0.85 | 6.66 ±0.20 | 5.14 ±0.73 | 1.27 ±0.20 | 0.59 ±0.08 | 0.22 ±0.02 | 0.11 ±0.02 | 0.11 ±0.02 |
| 41°39′47.22″ | 100°58′30.60″ | 14.39 ±0.55 | 8.35 ±0.43 | 5.13 ±0.63 | 1.98 ±0.44 | 0.52 ±0.03 | 0.23 ±0.01 | 0.09 ±0.01 | 0.12 ±0.02 |
| 42°00′49.37″ | 101°42′23.12″ | 10.55 ±0.44 | 8.62 ±0.60 | 1.68 ±0.17 | 0.70 ±0.17 | 0.33 ±0.03 | 0.25 ±0.03 | 0.08 ±0.01 | 0.09 ±0.02 |
| 41°57′49.08″ | 102°20′41.89″ | 6.45 ±0.36 | 3.73 ±0.27 | 1.74 ±0.37 | 0.97 ±0.25 | 0.31 ±0.05 | 0.28 ±0.02 | 0.04 ±0.01 | 0.09 ±0.01 |
| 41°43′02.80″ | 103°06′51.38″ | 14.99 ±0.39 | 8.03 ±0.40 | 4.27 ±0.16 | 2.99 ±0.45 | 0.57 ±0.06 | 0.26 ±0.03 | 0.11 ±0.01 | 0.20 ±0.03 |
| 41°21′26.33″ | 103°45′42.79″ | 6.86 ±0.37 | 4.68 ±0.32 | 1.17 ±0.25 | 0.74 ±0.14 | 0.29 ±0.02 | 0.27 ±0.03 | 0.03 ±0.01 | 0.08 ±0.01 |
| 40°52′33.64″ | 104°26′49.14″ | 14.6 ±0.84 | 5.26 ±0.62 | 6.3 ±0.93 | 3.01 ±0.51 | 0.50 ±0.06 | 0.20 ±0.02 | 0.18 ±0.02 | 0.19 ±0.03 |
| 40°47′37.30″ | 104°52′51.45″ | 9.14 ±1.64 | 2.49 ±0.11 | 3.66 ±0.97 | 2.69 ±0.86 | 0.40 ±0.05 | 0.20 ±0.03 | 0.10 ±0.02 | 0.18 ±0.06 |

| | | | | | | | | | |
|---|---|---|---|---|---|---|---|---|---|
| 40°43′44.48″ | 105°36′43.8″ | 7.22 ±0.86 | 3.65 ±0.61 | 1.85 ±0.31 | 1.23 ±0.17 | 0.35 ±0.05 | 0.30 ±0.04 | 0.09 ±0.02 | 0.14 ±0.02 |
| 40°41′025″ | 106°02′886″ | 9.97 ±1.77 | 2.14 ±0.11 | 4.68 ±0.54 | 3.96 ±0.59 | 0.59 ±0.10 | 0.23 ±0.04 | 0.24 ±0.03 | 0.26 ±0.03 |
| 41°27′02.23″ | 107°00′06.04″ | 19.94 ±3.4 | 1.21 ±0.19 | 7.69 ±1.25 | 9.72 ±1.89 | 1.19 ±0.13 | 0.18 ±0.02 | 0.47 ±0.05 | 0.57 ±0.06 |
| 41°47′46.44″ | 107°28′11.62″ | 10.19 ±0.49 | 0.88 ±0.17 | 4.24 ±0.25 | 3.96 ±0.41 | 0.93 ±0.03 | 0.21 ±0.03 | 0.40 ±0.03 | 0.36 ±0.02 |
| 41°49′42.97″ | 107°36′40.26″ | 10.17 ±0.64 | 0.77 ±0.04 | 3.75 ±0.25 | 4.15 ±0.36 | 1.03 ±0.08 | 0.20 ±0.03 | 0.42 ±0.04 | 0.46 ±0.04 |
| 41°51′57.84″ | 108°03′14.43″ | 7.71 ±0.33 | 1.27 ±0.12 | 2.69 ±0.17 | 2.79 ±0.20 | 0.76 ±0.07 | 0.23 ±0.03 | 0.26 ±0.03 | 0.37 ±0.03 |
| 41°54′52.38″ | 108°42′38.69″ | 10.8 ±0.60 | 1.17 ±0.04 | 4.14 ±0.22 | 4.36 ±0.30 | 1.28 ±0.09 | 0.23 ±0.03 | 0.53 ±0.02 | 0.61 ±0.03 |
| 42°09′46.07″ | 109°09′56.15″ | 8.48 ±0.43 | 0.96 ±0.08 | 2.88 ±0.20 | 3.83 ±0.29 | 0.94 ±0.07 | 0.18 ±0.03 | 0.31 ±0.02 | 0.47 ±0.03 |
| 42°24′54.72″ | 109°48′18.06″ | 7.01 ±0.43 | 0.60 ±0.03 | 2.93 ±0.20 | 2.58 ±0.26 | 1.13 ±0.08 | 0.24 ±0.03 | 0.44 ±0.03 | 0.49 ±0.05 |
| 42°37′26.24″ | 110°17′41.72″ | 6.89 ±0.54 | 0.89 ±0.04 | 2.62 ±0.34 | 2.51 ±0.30 | 0.90 ±0.05 | 0.30 ±0.03 | 0.30 ±0.03 | 0.38 ±0.02 |
| 42°55′54.74″ | 110°49′27.19″ | 4.79 ±0.17 | 0.56 ±0.04 | 1.79 ±0.15 | 1.82 ±0.11 | 0.75 ±0.06 | 0.25 ±0.04 | 0.26 ±0.02 | 0.32 ±0.02 |
| 43°08′49.69″ | 111°21′18.26″ | 8.56 ±0.90 | 0.78 ±0.06 | 3.16 ±0.35 | 3.33 ±0.35 | 0.99 ±0.08 | 0.22 ±0.01 | 0.41 ±0.04 | 0.51 ±0.03 |
| 43°22′54.51″ | 111°57′51.32″ | 14.27 ±0.32 | 1.03 ±0.05 | 5.86 ±0.14 | 6.20 ±0.29 | 1.07 ±0.02 | 0.17 ±0.01 | 0.43 ±0.01 | 0.56 ±0.01 |
| 43°38′05.12″ | 112°11′47.03″ | 4.36 ±0.40 | 0.65 ±0.06 | 1.07 ±0.22 | 1.72 ±0.07 | 0.58 ±0.06 | 0.21 ±0.02 | 0.15 ±0.02 | 0.26 ±0.01 |
| 43°42′25.70″ | 112°55′16.72″ | 9.04 ±1.03 | 0.60 ±0.04 | 3.51 ±0.55 | 3.80 ±0.48 | 1.04 ±0.02 | 0.18 ±0.01 | 0.40 ±0.02 | 0.48 ±0.02 |
| 43°49′08.29″ | 113°27′58.82″ | 6.38 ±0.49 | 0.98 ±0.06 | 2.32 ±0.17 | 2.11 ±0.15 | 0.75 ±0.06 | 0.20 ±0.02 | 0.29 ±0.02 | 0.33 ±0.02 |
| 43°50′58.62″ | 114°05′08.22″ | 9.90 ±0.43 | 1.11 ±0.09 | 3.68 ±0.31 | 3.83 ±0.15 | 1.38 ±0.04 | 0.24 ±0.01 | 0.48 ±0.04 | 0.59 ±0.02 |
| 43°58′46.01″ | 114°49′36.29″ | 15.63 ±2.46 | 1.27 ±0.29 | 6.75 ±0.83 | 6.67 ±1.08 | 1.89 ±0.11 | 0.20 ±0.03 | 0.79 ±0.08 | 0.86 ±0.05 |
| 43°55′33.55″ | 115°42′06.75″ | 13.57 ±1.03 | 1.08 ±0.12 | 5.80 ±0.48 | 5.72 ±0.45 | 1.78 ±0.13 | 0.20 ±0.01 | 0.71 ±0.05 | 0.80 ±0.06 |
| 44°13′17.46″ | 116°30′25.43″ | 16.52 ±0.23 | 1.03 ±0.09 | 8.23 ±0.34 | 6.59 ±0.11 | 1.97 ±0.02 | 0.23 ±0.01 | 0.92 ±0.03 | 0.84 ±0.01 |
| 44°27′59.70″ | 117°10′47.04″ | 21.04 ±2.11 | 2.13 ±0.18 | 9.97 ±0.92 | 8.68 ±0.97 | 2.39 ±0.24 | 0.24 ±0.02 | 1.08 ±0.09 | 1.08 ±0.10 |
| 44°39′58.27″ | 117°53′44.73″ | 29.75 ±1.29 | 2.99 ±0.35 | 13.4 ±0.59 | 12.4 ±0.37 | 3.15 ±0.10 | 0.32 ±0.03 | 1.39 ±0.06 | 1.49 ±0.05 |
| 44°59′23.89″ | 118°44′44.89″ | 20.49 ±1.39 | 3.71 ±0.43 | 9.19 ±0.97 | 5.88 ±0.40 | 2.10 ±0.14 | 0.34 ±0.04 | 0.94 ±0.09 | 0.69 ±0.05 |
| 45°25′36.47″ | 119°43′23.02″ | 30.44 ±1.56 | 4.04 ±0.74 | 13.76 ±0.63 | 9.60 ±0.41 | 2.84 ±0.11 | 0.32 ±0.04 | 1.21 ±0.05 | 1.01 ±0.04 |
| 46°22′37.77″ | 120°28′38.48″ | 43.06 ±3.25 | 7.51 ±1.38 | 20.66 ±1.56 | 15.36 ±1.04 | 3.89 ±0.28 | 0.57 ±0.09 | 1.57 ±0.12 | 1.55 ±0.10 |

| | | | | | | | | |
|---|---|---|---|---|---|---|---|---|---|
| 47°39′21.62″ | 119°17′57.40″ | 35.76±2.28 | 5.08±0.61 | 17.15±1.8 | 13.91±0.51 | 3.27±0.22 | 0.38±0.05 | 1.46±0.19 | 1.42±0.05 |
| 48°05′19.80″ | 118°27′20.04″ | 23.41±0.74 | 2.10±0.26 | 11.3±0.44 | 9.08±0.23 | 2.39±0.06 | 0.22±0.03 | 1.06±0.05 | 1.03±0.02 |
| 48°20′40.36″ | 117°58′46.17″ | 14.95±0.96 | 1.37±0.15 | 5.97±0.35 | 6.55±0.34 | 1.65±0.11 | 0.28±0.04 | 0.59±0.03 | 0.79±0.03 |
| 48°29′493″ | 117°09′716″ | 10.57±0.27 | 1.52±0.11 | 4.06±0.21 | 4.16±0.12 | 1.33±0.03 | 0.29±0.03 | 0.41±0.01 | 0.51±0.01 |
| 48°51′26.99″ | 116°53′36.00″ | 13.74±1.39 | 1.18±0.17 | 5.40±0.50 | 6.87±0.72 | 1.71±0.14 | 0.20±0.03 | 0.54±0.05 | 0.84±0.08 |
| 49°20′17.60″ | 117°05′28.24″ | 18.24±2.04 | 3.26±0.50 | 8.90±1.26 | 5.93±0.42 | 2.01±0.20 | 0.38±0.08 | 0.82±0.12 | 0.69±0.04 |
| 49°31′45.39″ | 118°00′35.84″ | 14.16±0.36 | 1.70±0.08 | 6.32±0.17 | 4.76±0.15 | 1.62±0.01 | 0.24±0.03 | 0.59±0.01 | 0.55±0.02 |
| 49°47′01.88″ | 118°32′00.47″ | 19.20±0.85 | 3.34±0.39 | 9.61±0.39 | 5.92±0.26 | 2.17±0.09 | 0.40±0.05 | 0.89±0.02 | 0.70±0.03 |
| 50°03′13.52″ | 119°16′57.97″ | 27.15±1.47 | 3.60±0.15 | 15.22±0.91 | 9.36±0.42 | 2.78±0.11 | 0.34±0.01 | 1.33±0.07 | 1.01±0.04 |
| 49°52′44.41″ | 119°59′36.93″ | 46.16±0.99 | 8.78±1.07 | 20.36±1.1 | 15.55±0.86 | 4.11±0.05 | 0.65±0.09 | 1.63±0.07 | 1.51±0.09 |
| 49°28′48.71″ | 119°40′55.44″ | 45.30±2.88 | 11.18±3.47 | 14.51±1.25 | 16.01±1.37 | 4.28±0.27 | 0.94±0.33 | 1.24±0.10 | 1.63±0.13 |
| 49°11′195″ | 120°21′451″ | 38.73±4.35 | 5.01±0.80 | 21.52±2.63 | 11.63±0.94 | 3.64±0.39 | 0.47±0.06 | 1.90±0.23 | 1.19±0.09 |
| 44°46′16.79″ | 123°23′04.99″ | 15.35±0.71 | 1.90±0.38 | 4.96±1.05 | 8.50±0.63 | 1.46±0.22 | 0.30±0.04 | 0.48±0.11 | 0.76±0.08 |


Table 3

| variables | unstandardized coefficients | | standardized coefficients | correlations | |
|---|---|---|---|---|---|
| | B | SE | Beta | partial | part |
| clay C concentration | 0.117 | 0.005 | 0.696 | 0.813 | 0.349 |
| clay content | 0.305 | 0.015 | 0.45 | 0.776 | 0.307 |
| silt content | 0.09 | 0.01 | 0.206 | 0.452 | 0.126 |
| silt C concentration | -0.034 | 0.008 | -0.122 | -0.258 | -0.067 |

Table 4

| variables | unstandardized coefficients | | standardized coefficients | correlations | |
|---|---|---|---|---|---|
| | B | SE | Beta | partial | part |
| BD | -0.246 | 0.088 | -0.1 | -0.164 | -0.042 |
| clay N concentration | 0.094 | 0.003 | 0.61 | 0.855 | 0.416 |
| clay content | 0.02 | 0.002 | 0.274 | 0.43 | 0.12 |
| sand content | -0.009 | 0.001 | -0.302 | -0.397 | -0.109 |
| sand N concentration | -0.033 | 0.015 | -0.052 | -0.124 | -0.031 |

Figure 1

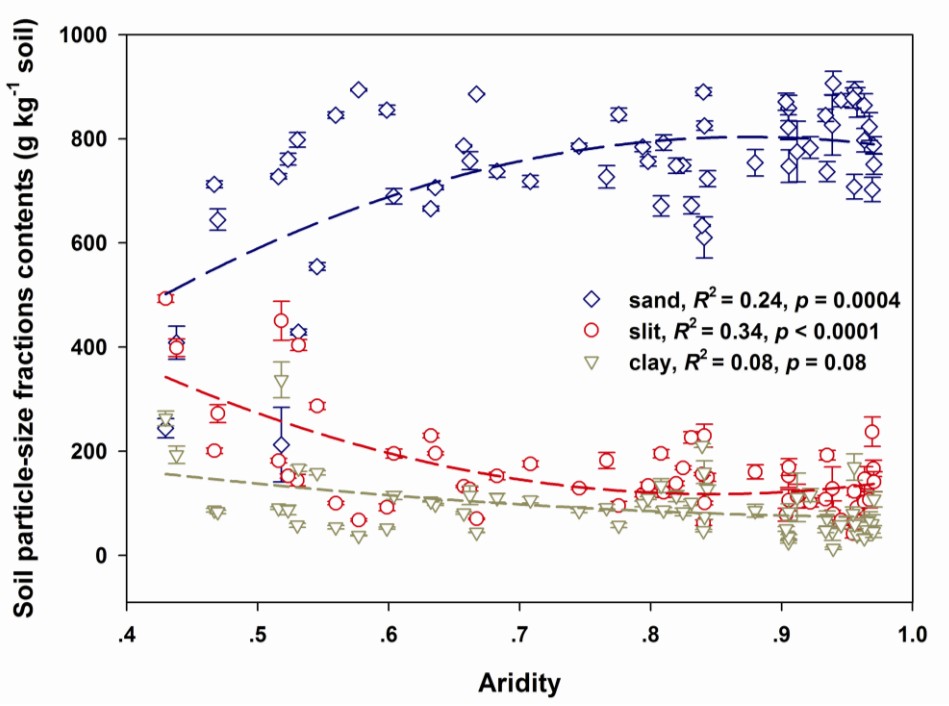


Figure 2

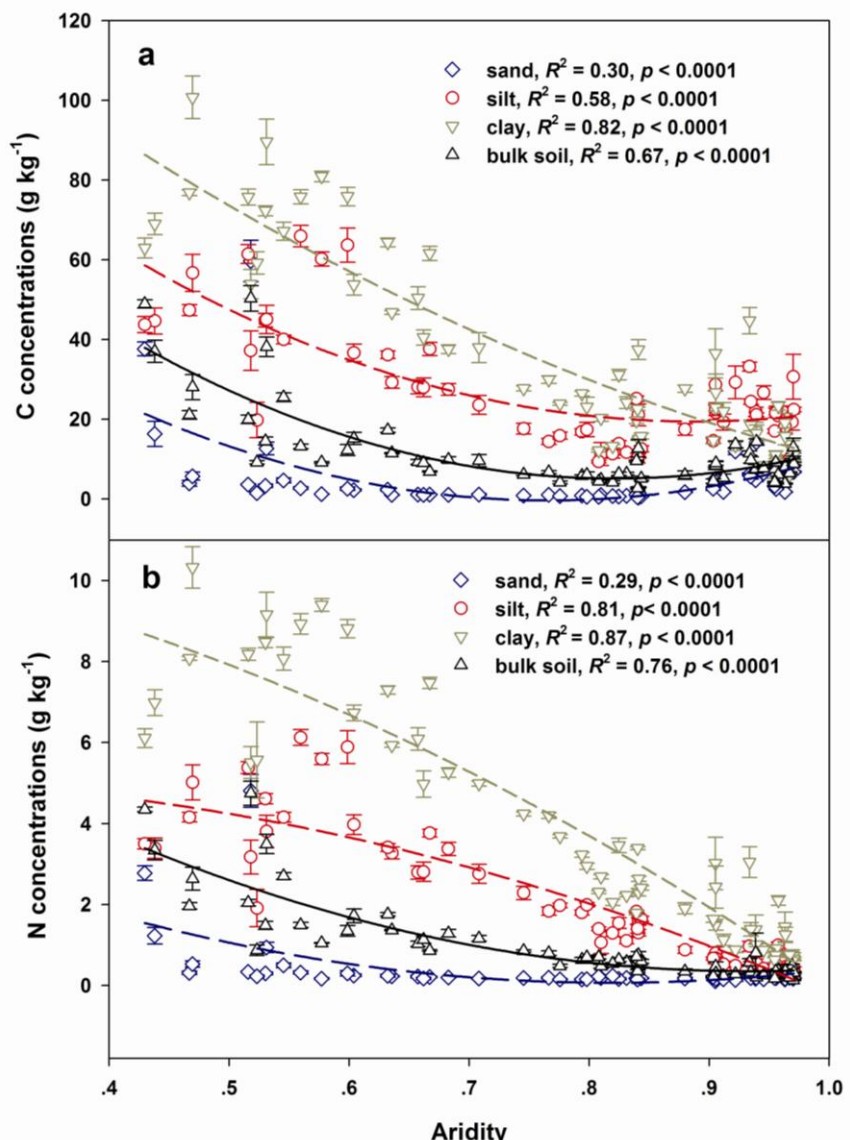

Figure 3

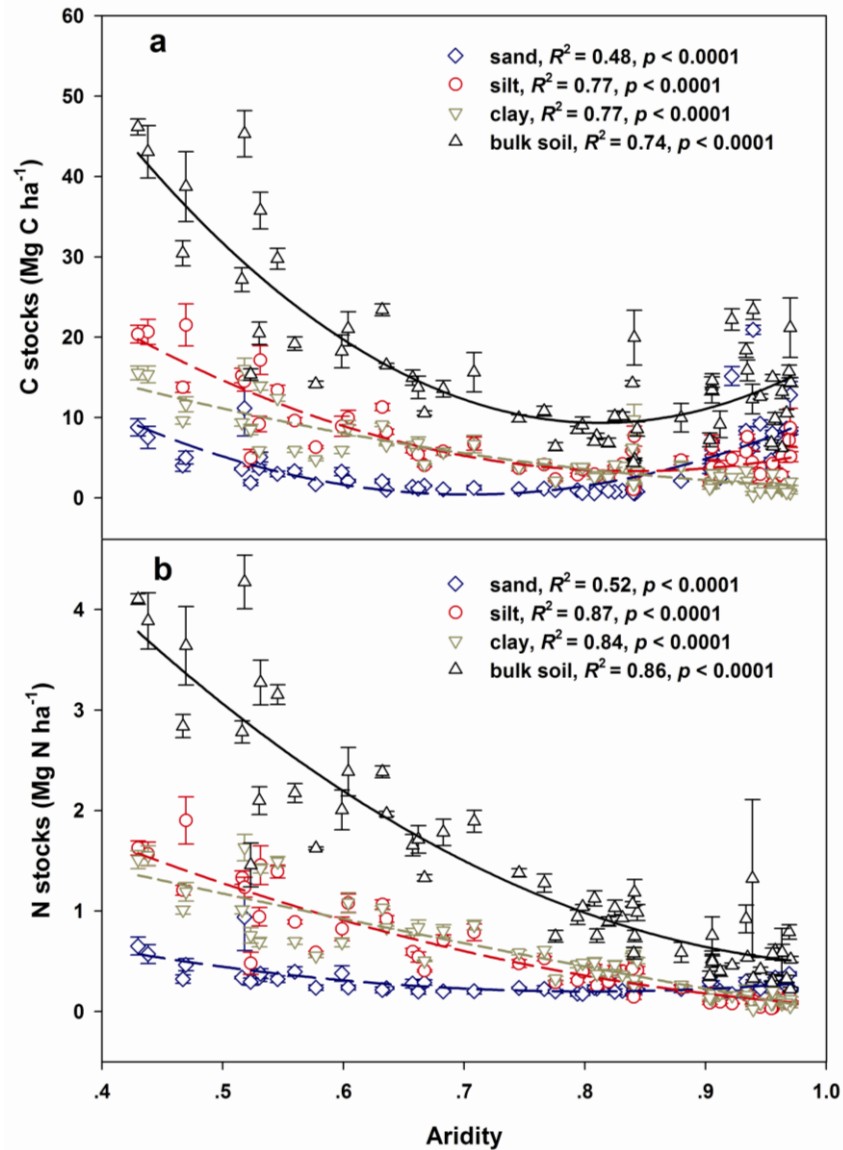

Figure 4

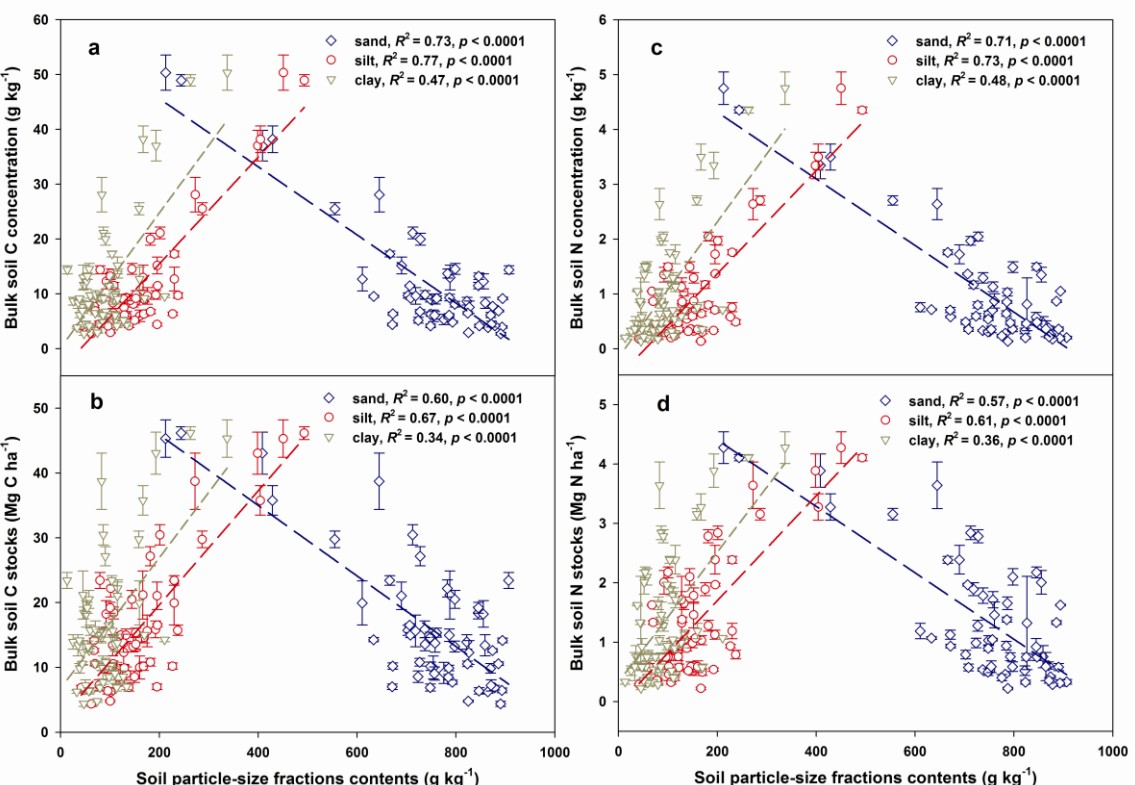

Figure 5

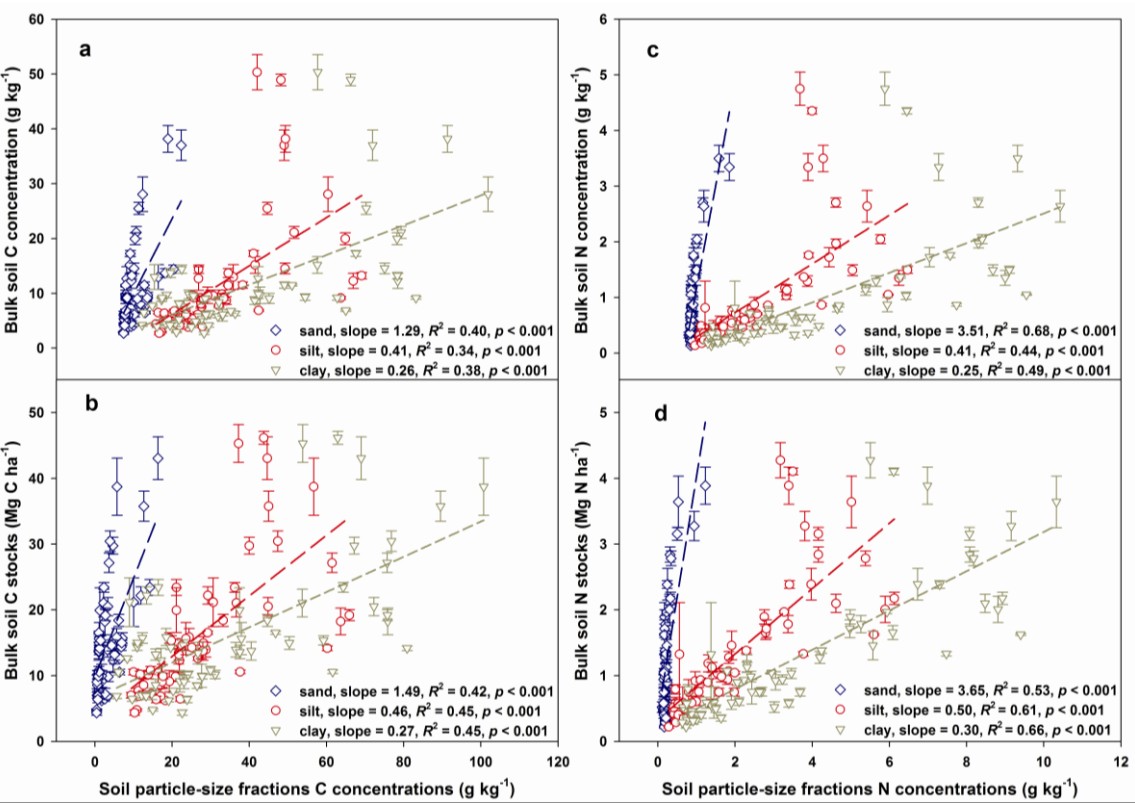