# Peer review of "Carbon and nitrogen contents in particle-size fractions of topsoil along a 3000 km aridity gradient in grasslands of northern China"

_Biogeosciences, 2015_

## Referee Comment (RC1) · Anonymous Referee #1 · 18 Mar 2016

This study analyzed the relationship between soil C and N, soil particle-size fractions and aridity in northern China. These authors found a negative relationship between soil C and N and aridity in both in bulk soils and fraction soils. It was also detected that sand fraction increased while silt and clay fraction decreased with increasing aridity. This study revealed that the diminishing effect of aridity on soil C and N was due to loss of fine soil particles as well as decrease of C and N in all soil particle-size fractions. The research had appropriate rational and study method which can provide helpful information on understanding C and N sequestration in drylands. Introduction and discussion needs to be improved to strengthen the significance of this study and data interpretation.

[Figure]

Pertaining to introduction, it is not clear on the importance of conducting research in this studying area. It would be useful to give more details regarding to environmental issues or concerns. For example, was this area experiencing increased aridity in recent years? Would local policy makers need suggestions on land or environmental management to mitigate the impacts of aridity? Introduction and discussion need to be expanded on the mechanism of soil C and N decrease due to aridity. Besides wind erosion and lower productivity, other aspects should also be considered, such as decline of soil water availability, soil microbial activity and diversity, aggregates formation, etc. Also, the paper focuses on particle-size fractions but doesn't have discussion on soil texture. Since large range of data was detected for sand (21.62-90.65%), silt (4.19-49.29%) and clay (1.36-33.7%) fraction, it would be good to discuss the changes in soil texture and corresponding soil characteristics.

Below are some specific comments on the paper:

Title:

Consider to include "grasslands" in the title

Abstract:

L16 "Soil composition" refers to soil water, air, minerals and organic matter. Should use "soil texture".

L19 Change "or" to "and/or"

Introduction:

L41- 44 Not clear why it is important to study C in grassland soils. Add a sentence indicating high percentage ($\sim$90%) of C in grassland ecosystems is stored in soil. Literature is also needed.

L62- 64 Reasons for less C and N in dryland other than decreased productivity is not stated here. Materials and Methods:

L123-125 Are soil types the same along this transect?

L126-128 Can 50m × 50m plot represent soils from each sampling area? Information regarding to soil homogeneity is needed.

Results:

L184- 192 Use a table to show the data for C and N concentration and stocks in bulk soil and soil fractions

L208- 223 Don't see discussion for these results.

Discussion

L231-232 Results for soil texture obtained from this study are missing. Would be good to include this data in discussion.

L235-238 Decreased microbial activity and diversity should also be discussed.

L246-248 How did land use affect soil type?

L268-271 Again, the discussion is not completed.

L278-290 Any citations to support this claim/result?

Conclusions:

Conclusions are missing in this paper.

---

## Referee Comment (RC2) · Anonymous Referee #2 · 16 Apr 2016

The manuscript entitled "Carbon and nitrogen stocks in particle-size fractions of topsoil along a 3000 km aridity gradient in northern China" examined the distribution of total C and N in bulk soil and different soil particle-size fractions along a 3000 km transect in arid and semi-arid grasslands of northern China, in order to relate the distribution of total C and N in bulk soil and their fractions with aridity and soil C and N stocks. The study is interesting and fits well the scope of the journal. The study deepened the understanding of the variations in content of soil particle-size fractions and their C and N concentrations with increasing aridity. Generally, the manuscript is well-structured and basically written in a concise style. And the results and conclusion is believable. However, some minor revisions should be made before publication in the journal. The

article is well written with good English and needs only some adjustments and clar-ifications to be acceptable for publication in Biogeosciences. In addition, I think that deleting less-important data in the results will improve the manuscript.

Specific comments: 1. Abstract, L18, change "whether such changes result from" to "whether such changes are resulted from"; 2. Introduction, L42, carbon (C); L46, nitrogen (N); 3. L48, extreme precipitation events; 4. L50, replace "Greater" with "Better"; 5. L71, change "so" to "and thus"; 6. L80. move the first paragraph to L62, before "Previous studies..."; 7. L94, the description of the selected transect should be move to the Study sites section. 8. Materials and methods: Since bulk density is an important factor to affect the calculated values of C or N stocks, the sampling methods should be provided in details. For example, how many cores were samples for each site? The methods of C or N stocks calculation should be described somewhere. 9. L140, The removal of visible debris should be performed immediately after sampling. As its hard to remove residues, especially roots, after the soils samples were air dried. 10. Results: L181, Soil C concentration; L182, N concentration; 11. L186, ($36.06\pm1.49$ g C kg-1 and $3.90\pm0.17$ g N kg-1, respectively); ($5.19\pm0.56$ g C kg-1 and $0.37\pm0.04$ g N kg-1, respectively); 12. From L210 to L220, Please consider that whether this section can be listed in a table, which would be much clearer. 13. Discussion: L236, which could further; 14. L239, "was" should be changed to "were"; 15. L288, change "looking at" to "considering"; 16. L301, aridity gradient were resulted.

---

## Author Comment (AC1) · 5 May 2016

Thank you for your comments on our manuscript entitled 'Carbon and nitrogen stocks in particle-size fractions of topsoil along a 3000 km aridity gradient in northern China'. We found the reviews are very useful, and we hope you will agree that our revised manuscript is substantially improved. Please find below our specific revisions in response to each comment.

Comments to the Author

This study analyzed the relationship between soil C and N, soil particle-size fractions and aridity in northern China. These authors found a negative relationship between

soil C and N and aridity in both in bulk soils and fraction soils. It was also detected that sand fraction increased while silt and clay fraction decreased with increasing aridity. This study revealed that the diminishing effect of aridity on soil C and N was due to loss of fine soil particles as well as decrease of C and N in all soil particle-size fractions. The research had appropriate rational and study method which can provide helpful information on understanding C and N sequestration in drylands. Introduction and discussion needs to be improved to strengthen the significance of this study and data interpretation.

Pertaining to introduction, it is not clear on the importance of conducting research in this studying area. It would be useful to give more details regarding to environmental issues or concerns. For example, was this area experiencing increased aridity in recent years? Would local policy makers need suggestions on land or environmental management to mitigate the impacts of aridity? Introduction and discussion need to be expanded on the mechanism of soil C and N decrease due to aridity. Besides wind erosion and lower productivity, other aspects should also be considered, such as decline of soil water availability, soil microbial activity and diversity, aggregates formation, etc. Also, the paper focuses on particle-size fractions but doesn't have discussion on soil texture. Since large range of data was detected for sand (21.62-90.65%), silt (4.19-49.29%) and clay (1.36-33.7%) fraction, it would be good to discuss the changes in soil texture and corresponding soil characteristics.

Response: Thank you for your positive comments. Following your suggestions, we added more details about this study area (please see lines 112-116, 128-130); we agreed with that besides wind erosion and lower productivity, other aspects that can influence soil C and N should also be considered. Please see our changes in lines 75-78, 290-294. We discussed the changes of soil texture (sand, silt and clay) with increasing aridity, please see lines 245-258.

Below are some specific comments on the paper:
Title: Consider to include "grasslands" in the title Response: Changed as suggested (the title, Line 2).

Abstract:

L16 "Soil composition" refers to soil water, air, minerals and organic matter. Should use "soil texture". Response: Changed as suggested (Line 14).

L19 Change "or" to "and/or" Response: Changed as suggested (Line 17).

Introduction:

L41- 44 Not clear why it is important to study C in grassland soils. Add a sentence indicating high percentage (∼90%) of C in grassland ecosystems is stored in soil. Literature is also needed. Response: Rewritten as suggested (Lines 40-42).

L62- 64 Reasons for less C and N in dryland other than decreased productivity is not stated here. Response: Please see our changes in Lines 75-78.

Materials and Methods:

L123-125 Are soil types the same along this transect? Response: Please see our changes in Lines 126-128.

L126-128 Can 50m × 50m plot represent soils from each sampling area? Information regarding to soil homogeneity is needed. Response: We do not have information regarding soil homogeneity in our research sites. As you can see from our results, the variation of soil characters in each site was relatively low, indicating that soil heterogeneity would not be high. Furthermore, we obtained samples from five different quadrats for each site. We believe the soil samples would well represent local soils from each sampling site.

Results:

L184- 192 Use a table to show the data for C and N concentration and stocks in bulk

soil and soil fractions Response: Thanks for your suggestion. Changed as suggested, please see Lines 521-534 and table 1 and table 2.

L208- 223 Don't see discussion for these results. Response: We added more contents in the discussion section. Please see Lines 324-345 in the discussion section.

Discussion

L231-232 Results for soil texture obtained from this study are missing. Would be good to include this data in discussion. Response: Thanks for your suggestion. Please see our changes in Lines 245-248.

L235-238 Decreased microbial activity and diversity should also be discussed. Response: Please see our changes in Lines 290-294.

L246-248 How did land use affect soil type? Response: Please see our changes in Lines 264-265

L268-271 Again, the discussion is not completed. Response: We added additional detail in this section. Please see these changes in Lines 290-294.

L278-290 Any citations to support this claim/result? Response: We did not find any document literatures related to this result and we discussed according to the data of this study.

Conclusions:

Conclusions are missing in this paper. Response: We added conclusions in the end of this manuscript. Please see Lines 360-371.

Many thanks for your constructive comments and suggestions, which have greatly helped us to improve this manuscript.

Please also note the supplement to this comment:
http://www.biogeosciences-discuss.net/bg-2015-651/bg-2015-651-AC1-

supplement.pdf

**Supplement:**

Table legend

**Table 1** Soil C and N concentrations in bulk soils and different soil particle-size fractions (sand, silt and clay) at 58 sampling sites in arid and semi-arid grasslands of northern China (Data are represented as means ± 1 SE, n=5)

**Table 2** Soil C and N stocks in bulk soils and different soil particle-size fractions (sand, silt and clay) at 58 sampling sites in arid and semi-arid grasslands of northern China (Data are represented as means ± 1 SE, n=5)

Table 1

| site | | C concentrations (g C kg$^{-1}$) | | | | N concentrations (g N kg$^{-1}$) | | | |
|---|---|---|---|---|---|---|---|---|---|
| latitude | longitude | bulk soil | sand | silt | clay | bulk soil | sand | silt | clay |
| 42°13′24.02″ | 87°22′37.83″ | 8.66±0.70 | 3.28±0.37 | 28.71±1.62 | 36.52±6.17 | 0.32±0.05 | 0.11±0.01 | 0.67±0.13 | 3.02±0.65 |
| 42°59′17.72″ | 90°25′32.02″ | 9.69±0.49 | 7.08±0.05 | 19.3±2.07 | 12.28±2.34 | 0.49±0.04 | 0.33±0.02 | 0.45±0.03 | 0.78±0.10 |
| 43°16′04.17″ | 91°15′35.87″ | 7.54±1.13 | 4.71±0.34 | 21.90±1.13 | 17.13±2.05 | 0.81±0.48 | 0.17±0.02 | 0.56±0.10 | 1.38±0.36 |
| 43°24′13.23″ | 91°54′43.13″ | 8.35±1.01 | 5.05±0.14 | 21.69±1.79 | 26.68±4.62 | 0.47±0.12 | 0.20±0.02 | 0.80±0.16 | 2.43±0.52 |
| 43°07′38.09″ | 92°48′45.88″ | 11.5±0.58 | 6.19±0.34 | 33.21±1.13 | 44.71±3.31 | 0.58±0.09 | 0.22±0.03 | 0.97±0.12 | 3.05±0.38 |
| 42°58′04.74″ | 93°27′34.27″ | 6.14±0.46 | 3.43±0.21 | 20.86±0.57 | 23.34±1.05 | 0.38±0.04 | 0.25±0.04 | 0.99±0.08 | 2.10±0.12 |
| 42°41′55.57″ | 93°58′53.51″ | 3.85±0.15 | 1.75±0.09 | 18.21±1.92 | 19.11±2.39 | 0.36±0.15 | 0.15±0.03 | 0.65±0.21 | 1.45±0.36 |
| 42°15′37.46″ | 94°16′44.10″ | 12.97±2.27 | 10.03±1.46 | 30.65±5.63 | 8.96±0.65 | 0.14±0.01 | 0.17±0.01 | 0.28±0.02 | 0.73±0.08 |
| 41°34′12.85″ | 95°17′50.04″ | 9.78±0.80 | 5.62±0.68 | 24.42±0.69 | 18.45±1.34 | 0.33±0.02 | 0.19±0.02 | 0.55±0.04 | 1.39±0.16 |
| 39°51′53.47″ | 98°39′20.30″ | 13.71±0.82 | 11.94±0.80 | 29.24±4.11 | 13.38±1.24 | 0.28±0.01 | 0.14±0.03 | 0.49±0.03 | 0.88±0.07 |
| 40°14′41.91″ | 99°20′35.82″ | 14.38±0.75 | 14.25±0.78 | 21.10±0.50 | 16.44±2.13 | 0.20±0.03 | 0.19±0.02 | 0.55±0.01 | 1.01±0.05 |
| 40°28′46.39″ | 99°51′59.92″ | 7.73±0.18 | 6.48±0.23 | 26.67±1.71 | 8.36±0.40 | 0.26±0.01 | 0.16±0.03 | 0.42±0.03 | 0.87±0.06 |
| 41°08′18.31″ | 100°27′38.72″ | 8.10±0.52 | 5.13±0.18 | 21.76±1.54 | 15.72±2.09 | 0.36±0.05 | 0.17±0.02 | 0.46±0.04 | 1.25±0.19 |
| 41°39′47.22″ | 100°58′30.60″ | 8.83±0.34 | 6.80±0.21 | 22.43±0.56 | 10.93±1.20 | 0.32±0.02 | 0.18±0.01 | 0.41±0.06 | 0.66±0.04 |
| 42°00′49.37″ | 101°42′23.12″ | 6.48±0.27 | 6.40±0.26 | 10.01±0.74 | 6.01±0.29 | 0.20±0.02 | 0.19±0.03 | 0.47±0.05 | 0.75±0.06 |
| 41°57′49.08″ | 102°20′41.89″ | 3.95±0.22 | 2.55±0.13 | 22.12±1.11 | 9.64±1.40 | 0.19±0.03 | 0.19±0.01 | 0.45±0.04 | 0.91±0.06 |
| 41°43′02.80″ | 103°06′51.38″ | 9.19±0.24 | 6.99±0.40 | 21.62±1.17 | 11.04±0.98 | 0.35±0.04 | 0.23±0.02 | 0.53±0.03 | 0.74±0.02 |
| 41°21′26.33″ | 103°45′42.79″ | 4.20±0.23 | 3.25±0.17 | 17.01±0.55 | 5.60±0.42 | 0.18±0.01 | 0.19±0.01 | 0.47±0.05 | 0.63±0.06 |
| 40°52′33.64″ | 104°26′49.14″ | 9.01±0.52 | 4.29±0.35 | 22.52±1.58 | 22.91±0.97 | 0.31±0.03 | 0.16±0.02 | 0.68±0.04 | 1.50±0.13 |
| 40°47′37.30″ | 104°52′51.45″ | 5.32±0.95 | 1.77±0.11 | 19.27±2.01 | 21.95±2.22 | 0.23±0.03 | 0.16±0.02 | 0.53±0.07 | 1.13±0.14 |
| 40°43′44.48″ | 105°36′43.8″ | 4.48±0.54 | 2.60±0.43 | 14.61±0.64 | 14.71±0.88 | 0.22±0.03 | 0.21±0.03 | 0.67±0.03 | 1.64±0.08 |

| | | | | | | | | | |
|---|---|---|---|---|---|---|---|---|---|
| 40°41′025″ | 106°02′886″ | 5.97 ±1.06 | 1.71 ±0.12 | 17.42 ±1.42 | 27.64 ±0.92 | 0.35 ±0.06 | 0.18 ±0.03 | 0.88 ±0.08 | 1.91 ±0.14 |
| 41°27′02.23″ | 107°00′06.04″ | 12.70 ±2.16 | 1.31 ±0.27 | 21.08 ±2.68 | 37.47 ±2.47 | 0.76 ±0.08 | 0.19 ±0.01 | 1.31 ±0.08 | 2.31 ±0.09 |
| 41°47′46.44″ | 107°28′11.62″ | 6.36 ±0.30 | 0.81 ±0.14 | 11.71 ±0.69 | 24.14 ±1.47 | 0.58 ±0.02 | 0.19 ±0.02 | 1.11 ±0.08 | 2.23 ±0.04 |
| 41°49′42.97″ | 107°36′40.26″ | 6.3 ±0.40 | 0.64 ±0.04 | 13.84 ±0.81 | 31.14 ±1.32 | 0.64 ±0.05 | 0.16 ±0.02 | 1.54 ±0.13 | 3.46 ±0.18 |
| 41°51′57.84″ | 108°03′14.43″ | 4.85 ±0.21 | 1.01 ±0.08 | 11.26 ±2.86 | 20.26 ±0.80 | 0.48 ±0.04 | 0.19 ±0.03 | 1.06 ±0.27 | 2.69 ±0.06 |
| 41°54′52.38″ | 108°42′38.69″ | 6.75 ±0.38 | 1.01 ±0.03 | 14.41 ±0.77 | 29.94 ±0.95 | 0.80 ±0.06 | 0.19 ±0.02 | 1.84 ±0.10 | 4.20 ±0.07 |
| 42°09′46.07″ | 109°09′56.15″ | 5.80 ±0.29 | 0.84 ±0.06 | 16.83 ±0.78 | 26.45 ±0.61 | 0.64 ±0.05 | 0.15 ±0.02 | 1.80 ±0.07 | 3.23 ±0.06 |
| 42°24′54.72″ | 109°48′18.06″ | 4.39 ±0.27 | 0.56 ±0.02 | 9.37 ±0.31 | 12.12 ±0.34 | 0.71 ±0.05 | 0.22 ±0.03 | 1.40 ±0.05 | 2.31 ±0.04 |
| 42°37′26.24″ | 110°17′41.72″ | 4.17 ±0.33 | 0.72 ±0.02 | 11.35 ±1.11 | 13.26 ±0.46 | 0.54 ±0.03 | 0.24 ±0.02 | 1.29 ±0.08 | 2.05 ±0.08 |
| 42°55′54.74″ | 110°49′27.19″ | 2.93 ±0.10 | 0.42 ±0.03 | 10.82 ±0.61 | 14.99 ±0.30 | 0.46 ±0.04 | 0.19 ±0.03 | 1.58 ±0.09 | 2.64 ±0.06 |
| 43°08′49.69″ | 111°21′18.26″ | 5.18 ±0.55 | 0.66 ±0.06 | 12.60 ±0.77 | 15.71 ±0.73 | 0.60 ±0.05 | 0.18 ±0.01 | 1.65 ±0.06 | 2.41 ±0.06 |
| 43°22′54.51″ | 111°57′51.32″ | 9.54 ±0.21 | 1.09 ±0.05 | 25.12 ±0.68 | 19.64 ±0.71 | 0.71 ±0.02 | 0.18 ±0.01 | 1.83 ±0.03 | 1.79 ±0.04 |
| 43°38′05.12″ | 112°11′47.03″ | 2.71 ±0.25 | 0.46 ±0.05 | 10.26 ±1.22 | 22.59 ±0.40 | 0.36 ±0.04 | 0.15 ±0.02 | 1.41 ±0.11 | 3.39 ±0.04 |
| 43°42′25.70″ | 112°55′16.72″ | 6.08 ±0.69 | 0.54 ±0.04 | 17.40 ±1.79 | 23.09 ±2.45 | 0.70 ±0.01 | 0.16 ±0.01 | 1.99 ±0.05 | 2.94 ±0.08 |
| 43°49′08.29″ | 113°27′58.82″ | 4.14 ±0.32 | 0.75 ±0.05 | 15.89 ±0.88 | 23.87 ±0.39 | 0.49 ±0.04 | 0.15 ±0.02 | 1.98 ±0.06 | 3.68 ±0.04 |
| 43°50′58.62″ | 114°05′08.22″ | 6.12 ±0.26 | 0.87 ±0.07 | 17.61 ±1.35 | 27.75 ±0.64 | 0.85 ±0.03 | 0.19 ±0.01 | 2.29 ±0.17 | 4.24 ±0.05 |
| 43°58′46.01″ | 114°49′36.29″ | 9.63 ±1.51 | 1.10 ±0.27 | 23.57 ±2.37 | 37.96 ±3.76 | 1.17 ±0.07 | 0.17 ±0.03 | 2.76 ±0.24 | 4.99 ±0.06 |
| 43°55′33.55″ | 115°42′06.75″ | 9.83 ±0.74 | 1.07 ±0.12 | 27.50 ±1.47 | 37.53 ±1.10 | 1.29 ±0.09 | 0.20 ±0.01 | 3.37 ±0.16 | 5.26 ±0.12 |
| 44°13′17.46″ | 116°30′25.43″ | 11.47 ±0.16 | 1.02 ±0.09 | 29.23 ±1.47 | 46.78 ±0.48 | 1.37 ±0.02 | 0.23 ±0.01 | 3.27 ±0.15 | 5.93 ±0.05 |
| 44°27′59.70″ | 117°10′47.04″ | 15.17 ±1.52 | 2.24 ±0.24 | 36.64 ±2.20 | 53.74 ±2.63 | 1.72 ±0.17 | 0.25 ±0.03 | 3.97 ±0.24 | 6.73 ±0.19 |
| 44°39′58.27″ | 117°53′44.73″ | 25.5 ±1.11 | 4.65 ±0.59 | 40.01 ±1.13 | 67.17 ±2.25 | 2.70 ±0.08 | 0.50 ±0.06 | 4.16 ±0.12 | 8.07 ±0.28 |
| 44°59′23.89″ | 118°44′44.89″ | 14.53 ±0.98 | 3.32 ±0.44 | 44.87 ±1.71 | 72.27 ±1.25 | 1.49 ±0.10 | 0.30 ±0.03 | 4.61 ±0.13 | 8.48 ±0.13 |
| 45°25′36.47″ | 119°43′23.02″ | 21.09 ±1.08 | 3.94 ±0.73 | 47.37 ±1.35 | 76.83 ±0.75 | 1.97 ±0.08 | 0.31 ±0.04 | 4.16 ±0.12 | 8.08 ±0.05 |
| 46°22′37.77″ | 120°28′38.48″ | 37.01 ±2.79 | 16.31 ±3.17 | 44.64 ±3.27 | 68.97 ±2.69 | 3.34 ±0.24 | 1.23 ±0.20 | 3.39 ±0.26 | 6.98 ±0.32 |
| 47°39′21.62″ | 119°17′57.40″ | 38.17 ±2.43 | 12.67 ±1.58 | 45.00 ±3.53 | 89.56 ±5.70 | 3.50 ±0.24 | 0.94 ±0.13 | 3.81 ±0.40 | 9.16 ±0.56 |

| | | | | | | | | | |
|---|---|---|---|---|---|---|---|---|---|
| 48°05′19.80″ | 118°27′20.04″ | 17.26 ±0.54 | 2.33 ±0.30 | 36.19 ±0.95 | 64.34 ±1.13 | 1.76 ±0.04 | 0.24 ±0.03 | 3.41 ±0.10 | 7.30 ±0.10 |
| 48°20′40.36″ | 117°58′46.17″ | 9.31 ±0.60 | 1.08 ±0.11 | 28.08 ±1.75 | 50.39 ±2.86 | 1.03 ±0.07 | 0.22 ±0.03 | 2.79 ±0.17 | 6.09 ±0.28 |
| 48°29′493″ | 117°09′716″ | 6.88 ±0.18 | 1.11 ±0.08 | 37.61 ±1.67 | 61.59 ±1.78 | 0.87 ±0.02 | 0.21 ±0.02 | 3.77 ±0.10 | 7.47 ±0.14 |
| 48°51′26.99″ | 116°53′36.00″ | 9.06 ±0.92 | 1.04 ±0.18 | 27.98 ±2.37 | 40.5 ±1.90 | 1.13 ±0.09 | 0.18 ±0.03 | 2.81 ±0.24 | 4.97 ±0.33 |
| 49°20′17.60″ | 117°05′28.24″ | 12.27 ±1.37 | 2.58 ±0.42 | 63.65 ±4.30 | 75.87 ±2.33 | 1.35 ±0.13 | 0.30 ±0.06 | 5.89 ±0.41 | 8.81 ±0.22 |
| 49°31′45.39″ | 118°00′35.84″ | 9.16 ±0.23 | 1.23 ±0.06 | 60.19 ±1.77 | 80.9 ±1.33 | 1.05 ±0.01 | 0.17 ±0.02 | 5.59 ±0.14 | 9.40 ±0.16 |
| 49°47′01.88″ | 118°32′00.47″ | 13.21 ±0.58 | 2.72 ±0.31 | 65.95 ±2.72 | 75.77 ±1.76 | 1.50 ±0.06 | 0.32 ±0.04 | 6.13 ±0.20 | 8.93 ±0.25 |
| 50°03′13.52″ | 119°16′57.97″ | 19.96 ±1.08 | 3.63 ±0.14 | 61.42 ±2.35 | 75.72 ±2.00 | 2.05 ±0.08 | 0.34 ±0.01 | 5.38 ±0.15 | 8.18 ±0.15 |
| 49°52′44.41″ | 119°59′36.93″ | 48.93 ±1.05 | 37.62 ±1.73 | 43.74 ±2.02 | 62.92 ±2.50 | 4.35 ±0.06 | 2.78 ±0.18 | 3.50 ±0.13 | 6.11 ±0.23 |
| 49°28′48.71″ | 119°40′55.44″ | 50.33 ±3.21 | 59.71 ±5.17 | 37.20 ±4.97 | 53.87 ±3.69 | 4.75 ±0.30 | 4.81 ±0.41 | 3.17 ±0.42 | 5.51 ±0.39 |
| 49°11′195″ | 120°21′451″ | 28.07 ±3.15 | 5.72 ±1.02 | 56.70 ±4.65 | 100.73 ±5.35 | 2.64 ±0.28 | 0.53 ±0.08 | 5.02 ±0.43 | 10.33 ±0.51 |
| 44°46′16.79″ | 123°23′04.99″ | 9.17 ±0.43 | 1.48 ±0.28 | 19.78 ±4.43 | 59.23 ±2.83 | 0.87 ±0.13 | 0.23 ±0.03 | 1.91 ±0.46 | 5.57 ±0.94 |

Table 2

| site | | C stocks (Mg C ha$^{-1}$) | | | | N stocks (Mg N ha$^{-1}$) | | | |
|---|---|---|---|---|---|---|---|---|---|
| latitude | longitude | bulk soil | sand | silt | clay | bulk soil | sand | silt | clay |
| 42°13′24.02″ | 87°22′37.83″ | 13.93±1.12 | 4.32±0.50 | 6.96±0.87 | 1.47±0.21 | 0.52±0.08 | 0.15±0.02 | 0.15±0.02 | 0.12±0.02 |
| 42°59′17.72″ | 90°25′32.02″ | 15.71±0.79 | 8.07±0.31 | 7.19±0.77 | 1.18±0.40 | 0.79±0.07 | 0.37±0.02 | 0.17±0.02 | 0.08±0.02 |
| 43°16′04.17″ | 91°15′35.87″ | 12.3±1.84 | 6.29±0.53 | 4.47±1.37 | 1.23±0.42 | 1.32±0.79 | 0.23±0.04 | 0.11±0.04 | 0.09±0.02 |
| 43°24′13.23″ | 91°54′43.13″ | 13.43±1.63 | 6.96±0.18 | 3.82±0.96 | 1.61±0.43 | 0.76±0.19 | 0.28±0.03 | 0.15±0.05 | 0.15±0.05 |
| 43°07′38.09″ | 92°48′45.88″ | 18.4±0.92 | 8.37±0.51 | 5.72±0.46 | 3.40±0.37 | 0.92±0.14 | 0.30±0.04 | 0.16±0.02 | 0.23±0.03 |
| 42°58′04.74″ | 93°27′34.27″ | 9.85±0.74 | 4.80±0.37 | 3.07±0.67 | 1.41±0.21 | 0.61±0.06 | 0.34±0.05 | 0.14±0.03 | 0.13±0.02 |
| 42°41′55.57″ | 93°58′53.51″ | 6.24±0.24 | 2.46±0.17 | 2.82±0.47 | 1.04±0.18 | 0.59±0.24 | 0.22±0.05 | 0.10±0.03 | 0.08±0.02 |
| 42°15′37.46″ | 94°16′44.10″ | 21.16±3.70 | 12.80±1.67 | 8.71±2.43 | 0.67±0.17 | 0.22±0.02 | 0.21±0.02 | 0.08±0.01 | 0.05±0.01 |
| 41°34′12.85″ | 95°17′50.04″ | 15.86±1.30 | 6.67±0.73 | 7.60±0.33 | 2.08±0.36 | 0.54±0.03 | 0.22±0.02 | 0.17±0.01 | 0.16±0.04 |
| 39°51′53.47″ | 98°39′20.30″ | 22.19±1.33 | 15.15±1.17 | 4.86±0.95 | 2.54±0.45 | 0.46±0.02 | 0.17±0.04 | 0.08±0.00 | 0.16±0.01 |
| 40°14′41.91″ | 99°20′35.82″ | 23.41±1.21 | 20.91±0.57 | 2.82±0.97 | 0.34±0.04 | 0.33±0.04 | 0.28±0.03 | 0.07±0.02 | 0.02±0.00 |
| 40°28′46.39″ | 99°51′59.92″ | 12.61±0.29 | 9.25±0.45 | 2.95±0.34 | 0.78±0.08 | 0.42±0.02 | 0.23±0.04 | 0.05±0.01 | 0.08±0.01 |
| 41°08′18.31″ | 100°27′38.72″ | 13.21±0.85 | 6.66±0.20 | 5.14±0.73 | 1.27±0.20 | 0.59±0.08 | 0.22±0.02 | 0.11±0.02 | 0.11±0.02 |
| 41°39′47.22″ | 100°58′30.60″ | 14.39±0.55 | 8.35±0.43 | 5.13±0.63 | 1.98±0.44 | 0.52±0.03 | 0.23±0.01 | 0.09±0.01 | 0.12±0.02 |
| 42°00′49.37″ | 101°42′23.12″ | 10.55±0.44 | 8.62±0.60 | 1.68±0.17 | 0.70±0.17 | 0.33±0.03 | 0.25±0.03 | 0.08±0.01 | 0.09±0.02 |
| 41°57′49.08″ | 102°20′41.89″ | 6.45±0.36 | 3.73±0.27 | 1.74±0.37 | 0.97±0.25 | 0.31±0.05 | 0.28±0.02 | 0.04±0.01 | 0.09±0.01 |
| 41°43′02.80″ | 103°06′51.38″ | 14.99±0.39 | 8.03±0.40 | 4.27±0.16 | 2.99±0.45 | 0.57±0.06 | 0.26±0.03 | 0.11±0.01 | 0.20±0.03 |
| 41°21′26.33″ | 103°45′42.79″ | 6.86±0.37 | 4.68±0.32 | 1.17±0.25 | 0.74±0.14 | 0.29±0.02 | 0.27±0.03 | 0.03±0.01 | 0.08±0.01 |
| 40°52′33.64″ | 104°26′49.14″ | 14.6±0.84 | 5.26±0.62 | 6.3±0.93 | 3.01±0.51 | 0.50±0.06 | 0.20±0.02 | 0.18±0.02 | 0.19±0.03 |
| 40°47′37.30″ | 104°52′51.45″ | 9.14±1.64 | 2.49±0.11 | 3.66±0.97 | 2.69±0.86 | 0.40±0.05 | 0.20±0.03 | 0.10±0.02 | 0.18±0.06 |

| | | | | | | | | | | |
|---|---|---|---|---|---|---|---|---|---|---|
| 40°43′44.48″ | 105°36′43.8″ | 7.22±0.86 | 3.65±0.61 | 1.85±0.31 | 1.23±0.17 | 0.35±0.05 | 0.30±0.04 | 0.09±0.02 | 0.14±0.02 |
| 40°41′025″ | 106°02′886″ | 9.97±1.77 | 2.14±0.11 | 4.68±0.54 | 3.96±0.59 | 0.59±0.10 | 0.23±0.04 | 0.24±0.03 | 0.26±0.03 |
| 41°27′02.23″ | 107°00′06.04″ | 19.94±3.4 | 1.21±0.19 | 7.69±1.25 | 9.72±1.89 | 1.19±0.13 | 0.18±0.02 | 0.47±0.05 | 0.57±0.06 |
| 41°47′46.44″ | 107°28′11.62″ | 10.19±0.49 | 0.88±0.17 | 4.24±0.25 | 3.96±0.41 | 0.93±0.03 | 0.21±0.03 | 0.40±0.03 | 0.36±0.02 |
| 41°49′42.97″ | 107°36′40.26″ | 10.17±0.64 | 0.77±0.04 | 3.75±0.25 | 4.15±0.36 | 1.03±0.08 | 0.20±0.03 | 0.42±0.04 | 0.46±0.04 |
| 41°51′57.84″ | 108°03′14.43″ | 7.71±0.33 | 1.27±0.12 | 2.69±0.17 | 2.79±0.20 | 0.76±0.07 | 0.23±0.03 | 0.26±0.03 | 0.37±0.03 |
| 41°54′52.38″ | 108°42′38.69″ | 10.8±0.60 | 1.17±0.04 | 4.14±0.22 | 4.36±0.30 | 1.28±0.09 | 0.23±0.03 | 0.53±0.02 | 0.61±0.03 |
| 42°09′46.07″ | 109°09′56.15″ | 8.48±0.43 | 0.96±0.08 | 2.88±0.20 | 3.83±0.29 | 0.94±0.07 | 0.18±0.03 | 0.31±0.02 | 0.47±0.03 |
| 42°24′54.72″ | 109°48′18.06″ | 7.01±0.43 | 0.60±0.03 | 2.93±0.20 | 2.58±0.26 | 1.13±0.08 | 0.24±0.03 | 0.44±0.03 | 0.49±0.05 |
| 42°37′26.24″ | 110°17′41.72″ | 6.89±0.54 | 0.89±0.04 | 2.62±0.34 | 2.51±0.30 | 0.90±0.05 | 0.30±0.03 | 0.30±0.03 | 0.38±0.02 |
| 42°55′54.74″ | 110°49′27.19″ | 4.79±0.17 | 0.56±0.04 | 1.79±0.15 | 1.82±0.11 | 0.75±0.06 | 0.25±0.04 | 0.26±0.02 | 0.32±0.02 |
| 43°08′49.69″ | 111°21′18.26″ | 8.56±0.90 | 0.78±0.06 | 3.16±0.35 | 3.33±0.35 | 0.99±0.08 | 0.22±0.01 | 0.41±0.04 | 0.51±0.03 |
| 43°22′54.51″ | 111°57′51.32″ | 14.27±0.32 | 1.03±0.05 | 5.86±0.14 | 6.20±0.29 | 1.07±0.02 | 0.17±0.01 | 0.43±0.01 | 0.56±0.01 |
| 43°38′05.12″ | 112°11′47.03″ | 4.36±0.40 | 0.65±0.06 | 1.07±0.22 | 1.72±0.07 | 0.58±0.06 | 0.21±0.02 | 0.15±0.02 | 0.26±0.01 |
| 43°42′25.70″ | 112°55′16.72″ | 9.04±1.03 | 0.60±0.04 | 3.51±0.55 | 3.80±0.48 | 1.04±0.02 | 0.18±0.01 | 0.40±0.02 | 0.48±0.02 |
| 43°49′08.29″ | 113°27′58.82″ | 6.38±0.49 | 0.98±0.06 | 2.32±0.17 | 2.11±0.15 | 0.75±0.06 | 0.20±0.02 | 0.29±0.02 | 0.33±0.02 |
| 43°50′58.62″ | 114°05′08.22″ | 9.90±0.43 | 1.11±0.09 | 3.68±0.31 | 3.83±0.15 | 1.38±0.04 | 0.24±0.01 | 0.48±0.04 | 0.59±0.02 |
| 43°58′46.01″ | 114°49′36.29″ | 15.63±2.46 | 1.27±0.29 | 6.75±0.83 | 6.67±1.08 | 1.89±0.11 | 0.20±0.03 | 0.79±0.08 | 0.86±0.05 |
| 43°55′33.55″ | 115°42′06.75″ | 13.57±1.03 | 1.08±0.12 | 5.80±0.48 | 5.72±0.45 | 1.78±0.13 | 0.20±0.01 | 0.71±0.05 | 0.80±0.06 |
| 44°13′17.46″ | 116°30′25.43″ | 16.52±0.23 | 1.03±0.09 | 8.23±0.34 | 6.59±0.11 | 1.97±0.02 | 0.23±0.01 | 0.92±0.03 | 0.84±0.01 |
| 44°27′59.70″ | 117°10′47.04″ | 21.04±2.11 | 2.13±0.18 | 9.97±0.92 | 8.68±0.97 | 2.39±0.24 | 0.24±0.02 | 1.08±0.09 | 1.08±0.10 |
| 44°39′58.27″ | 117°53′44.73″ | 29.75±1.29 | 2.99±0.35 | 13.4±0.59 | 12.4±0.37 | 3.15±0.10 | 0.32±0.03 | 1.39±0.06 | 1.49±0.05 |
| 44°59′23.89″ | 118°44′44.89″ | 20.49±1.39 | 3.71±0.43 | 9.19±0.97 | 5.88±0.40 | 2.10±0.14 | 0.34±0.04 | 0.94±0.09 | 0.69±0.05 |
| 45°25′36.47″ | 119°43′23.02″ | 30.44±1.56 | 4.04±0.74 | 13.76±0.63 | 9.60±0.41 | 2.84±0.11 | 0.32±0.04 | 1.21±0.05 | 1.01±0.04 |
| 46°22′37.77″ | 120°28′38.48″ | 43.06±3.25 | 7.51±1.38 | 20.66±1.56 | 15.36±1.04 | 3.89±0.28 | 0.57±0.09 | 1.57±0.12 | 1.55±0.10 |

| | | | | | | | | | |
|---|---|---|---|---|---|---|---|---|---|
| 47°39′21.62″ | 119°17′57.40″ | 35.76±2.28 | 5.08±0.61 | 17.15±1.8 | 13.91±0.51 | 3.27±0.22 | 0.38±0.05 | 1.46±0.19 | 1.42±0.05 |
| 48°05′19.80″ | 118°27′20.04″ | 23.41±0.74 | 2.10±0.26 | 11.3±0.44 | 9.08±0.23 | 2.39±0.06 | 0.22±0.03 | 1.06±0.05 | 1.03±0.02 |
| 48°20′40.36″ | 117°58′46.17″ | 14.95±0.96 | 1.37±0.15 | 5.97±0.35 | 6.55±0.34 | 1.65±0.11 | 0.28±0.04 | 0.59±0.03 | 0.79±0.03 |
| 48°29′493″ | 117°09′716″ | 10.57±0.27 | 1.52±0.11 | 4.06±0.21 | 4.16±0.12 | 1.33±0.03 | 0.29±0.03 | 0.41±0.01 | 0.51±0.01 |
| 48°51′26.99″ | 116°53′36.00″ | 13.74±1.39 | 1.18±0.17 | 5.40±0.50 | 6.87±0.72 | 1.71±0.14 | 0.20±0.03 | 0.54±0.05 | 0.84±0.08 |
| 49°20′17.60″ | 117°05′28.24″ | 18.24±2.04 | 3.26±0.50 | 8.90±1.26 | 5.93±0.42 | 2.01±0.20 | 0.38±0.08 | 0.82±0.12 | 0.69±0.04 |
| 49°31′45.39″ | 118°00′35.84″ | 14.16±0.36 | 1.70±0.08 | 6.32±0.17 | 4.76±0.15 | 1.62±0.01 | 0.24±0.03 | 0.59±0.01 | 0.55±0.02 |
| 49°47′01.88″ | 118°32′00.47″ | 19.20±0.85 | 3.34±0.39 | 9.61±0.39 | 5.92±0.26 | 2.17±0.09 | 0.40±0.05 | 0.89±0.02 | 0.70±0.03 |
| 50°03′13.52″ | 119°16′57.97″ | 27.15±1.47 | 3.60±0.15 | 15.22±0.91 | 9.36±0.42 | 2.78±0.11 | 0.34±0.01 | 1.33±0.07 | 1.01±0.04 |
| 49°52′44.41″ | 119°59′36.93″ | 46.16±0.99 | 8.78±1.07 | 20.36±1.1 | 15.55±0.86 | 4.11±0.05 | 0.65±0.09 | 1.63±0.07 | 1.51±0.09 |
| 49°28′48.71″ | 119°40′55.44″ | 45.30±2.88 | 11.18±3.47 | 14.51±1.25 | 16.01±1.37 | 4.28±0.27 | 0.94±0.33 | 1.24±0.10 | 1.63±0.13 |
| 49°11′195″ | 120°21′451″ | 38.73±4.35 | 5.01±0.80 | 21.52±2.63 | 11.63±0.94 | 3.64±0.39 | 0.47±0.06 | 1.90±0.23 | 1.19±0.09 |
| 44°46′16.79″ | 123°23′04.99″ | 15.35±0.71 | 1.90±0.38 | 4.96±1.05 | 8.50±0.63 | 1.46±0.22 | 0.30±0.04 | 0.48±0.11 | 0.76±0.08 |

---

## Author Comment (AC2) · 5 May 2016

Thank you for your comments on our manuscript entitled 'Carbon and nitrogen stocks in particle-size fractions of topsoil along a 3000 km aridity gradient in northern China'. We found the reviews are very useful, and we hope you will agree that our revised manuscript is substantially improved. Please find below our specific revisions in response to each comment.

Comments to the Author

The manuscript entitled "Carbon and nitrogen stocks in particle-size fractions of topsoil along a 3000 km aridity gradient in northern China" examined the distribution of total

C and N in bulk soil and different soil particle-size fractions along a 3000 km transect in arid and semi-arid grasslands of northern China, in order to relate the distribution of total C and N in bulk soil and their fractions with aridity and soil C and N stocks. The study is interesting and fits well the scope of the journal. The study deepened the understanding of the variations in content of soil particle-size fractions and their C and N concentrations with increasing aridity. Generally, the manuscript is well-structured and basically written in a concise style. And the results and conclusion is believable. However, some minor revisions should be made before publication in the journal. The article is well written with good English and needs only some adjustments and clarifications to be acceptable for publication in Biogeosciences. In addition, I think that deleting less-important data in the results will improve the manuscript.

Specific comments:

1. Abstract, L18, change "whether such changes result from" to "whether such changes are resulted from". Response: Thank you for your positive comments. Changed as suggested (Line 16).

2. Introduction, L42, carbon (C); L46, nitrogen (N). Response: Changed as suggested (Line 40, 45)

3. L48, extreme precipitation events. Response: Changed as suggested (Line 47)

4. L50, replace "Greater" with "Better". Response: Changed as suggested (Line 49)

5. L71, change "so" to "and thus". Response: Changed as suggested (Line 84)

6. L80. move the first paragraph to L62, before "Previous studies...". Response: Thank you for your suggestion. Changed as suggested (Line 62-72)

7. L94, the description of the selected transect should be move to the Study sites section. Response: Thank you for your suggestion. We moved these descriptions to the study sites section. Please see lines 112-113.

8. Materials and methods: Since bulk density is an important factor to affect the calculated values of C or N stocks, the sampling methods should be provided in details. For example, how many cores were samples for each site? The methods of C or N stocks calculation should be described somewhere. Response: Thank you for your suggestion. Changed as suggested, please see line 140 and lines 164-175.

9. L140, The removal of visible debris should be performed immediately after sampling. As its hard to remove residues, especially roots, after the soils samples were air dried. Response: We agree with your advice. In fact, most of the visible debris was removed after sampling. But when we put the soils in deionized water, there were remain some floating debris and we also removed them. We now explain this in the description. Please see lines 146-147.

10. Results: L181, Soil C concentration; L182, N concentration. Response: Changed as suggested (Lines 199, 200)

11. L186, (36.06±1.49 g C kg-1 and 3.90±0.17 g N kg-1, respectively); (5.19±0.56 g C kg-1 and 0.37±0.04 g N kg-1, respectively). Response: Changed as suggested (Lines 204-206)

12. From L210 to L220, Please consider that whether this section can be listed in a table, which would be much clearer. Response: Thank you for your suggestion. We listed the data of this section in two tables (please see table 3, 4; Line 230 and 235).

13. Discussion: L236, which could further. Response: We deleted this sentence and discussed this in Lines 290-294.

14. L239, "was" should be changed to "were". Response: Changed as suggested (Line 256)

15. L288, change "looking at" to "considering". Response: Changed as suggested (Line 313)

16. L301, aridity gradient were resulted. Response: Changed as suggested (Line 325)

Many thanks for your constructive comments and suggestions, which have greatly helped us to improve this manuscript.

Please also note the supplement to this comment:
http://www.biogeosciences-discuss.net/bg-2015-651/bg-2015-651-AC2-supplement.pdf

**Supplement:**

**Table 3** Results of the multiple regressions refer to the final accepted model which just included

the effects of the significant variables for C stocks in bulk soils of arid and semi-arid grasslands

**Table 4** Results of the multiple regressions refer to the final accepted model which just included

the effects of the significant variables for N stocks in bulk soils of arid and semi-arid grasslands

Table 3

| variables | unstandardized coefficients | | standardized coefficients | correlations | |
|---|---|---|---|---|---|
| | B | SE | Beta | partial | part |
| clay C concentration | 0.117 | 0.005 | 0.696 | 0.813 | 0.349 |
| clay content | 0.305 | 0.015 | 0.45 | 0.776 | 0.307 |
| silt content | 0.09 | 0.01 | 0.206 | 0.452 | 0.126 |
| silt C concentration | -0.034 | 0.008 | -0.122 | -0.258 | -0.067 |

Table 4

| variables | unstandardized coefficients | | standardized coefficients | correlations | |
|---|---|---|---|---|---|
| | B | SE | Beta | partial | part |
| BD | -0.246 | 0.088 | -0.1 | -0.164 | -0.042 |
| clay N concentration | 0.094 | 0.003 | 0.61 | 0.855 | 0.416 |
| clay content | 0.02 | 0.002 | 0.274 | 0.43 | 0.12 |
| sand content | -0.009 | 0.001 | -0.302 | -0.397 | -0.109 |
| sand N concentration | -0.033 | 0.015 | -0.052 | -0.124 | -0.031 |

---

## Author Response (AR1)

Dear Editor,

Thank you for your positive decision on our manuscript entitled "Carbon and nitrogen stocks in particle-size fractions of topsoil along a 3000 km aridity gradient in northern China" (bg-2015-651). We found your suggestions are very useful, and we hope you will agree that our revised manuscript is substantially improved.

Please find below our specific revisions in response to each suggestion, and please let us know if you have any questions regarding these changes.

We look forward to your reply.

On behalf of the coauthors,

Yours sincerely,

Xiaotao Lü

**Our responses to the editor:**

Comments to the authors:

(1) The title. Carbon and nitrogen storage is an appropriate term for the bulk soil as a whole. But it seems for me that carbon and nitrogen content is often used when it refers to particle-size fraction. Therefore, I am not sure whether carbon and nitrogen stock is better than the term storage. pls consult soil chemist for this matter. I think the concentration or content is better than stock.

**Response:** Thank you. Following your suggestion, we decide to use 'content' in the title. Please see our changes in the title (line 1) and in the abstract (lines 15, 22, 23, 25, 27, 29, 32).

(2) The mechanisms behind the decreasing carbon and nitrogen content along increasing aridity. This observation might be a net result of lower above-ground biomass under higher aridity over a geological timescale. It could also result in higher content of sand component in temperate steppe.

**Response:** Thank you for this great suggestion. We added more content about the changes of biomass and plant coverage along the transect in the discussion section. Please see our changes in lines 324-328.

**Many thanks for your suggestions, which have helped us improve this manuscript.**

**Marked-up manuscript version:**

[revised manuscript text omitted]

Figure 2

[Figure]

Figure 3

[Figure]

Figure 4

[Figure]

Figure 5

[Figure]